

Title: Evaluation of a Hierarchical Agglomerative Clustering Method Applied to WIBS
Laboratory Data for Improved Discrimination of Biological Particles by Comparing Data
Preparation Techniques
NICOLE SAVAGE[1#], J Alex Huffman[1]
[1] *University of Denver, Department of Chemistry and Biochemistry, Denver, USA*
[#] *Now at Aerosol Devices, Inc.*
*Correspondence to:* J. Alex Huffman (alex.huffman@du.edu)
Running Title: Evaluation of clustering applied to WIBS bioaerosol data
Keywords: Clustering, Thresholding, Ward's linkage, Bioaerosols, Fluorescence, Laboratory
characterization

**Abstract**
Hierarchical agglomerative clustering (HAC) analysis has been successfully applied to
several sets of ambient data (e.g. Crawford et al., 2015; Robinson et al., 2013) and with respect
to standardized particles in the laboratory environment (Ruske et al., 2017). Here we show for
the first time a systematic application of HAC to a comprehensive set of laboratory data
collected using the wideband integrated bioaerosol sensor (WIBS-4A) (Savage et al., 2017). The
impact of particle ratio on HAC results was investigated, showing that clustering quality can
vary dramatically as a function of ratio. Six strategies for particle pre-processing were also
compared, concluding that using raw fluorescence intensity (without normalizing to particle size)
and inputting all data in logarithmic bins consistently produced the highest quality results. A
total of 23 one-on-one matchups of individual particles types were investigated. Results showed
cluster misclassification of <15% for 12 of 17 analytical experiments using one biological and
one non-biological particle type each. Inputting fluorescence data using a baseline + 3σ threshold
produced lower misclassification than when inputting either all particles (without fluorescence
threshold) or a baseline + 9σ threshold. Lastly, six synthetic mixtures of four to seven
components were analyzed. These results show that a range of 12-24% of fungal clusters were
consistently misclassified by inclusion of a mixture of non-biological materials, whereas bacteria
and diesel soot were each able to be separated with nearly 100% efficiency. The study gives
significant support to the application of clustering analysis to data from commercial UV-LIF
instruments being commonly used for bioaerosol research across the globe and provides practical
tools that will improve clustering results within scientific studies as a part of diverse research
disciplines.





## 1. Introduction

Particles of biological origin, or bioaerosols, make up a substantial fraction of atmospheric aerosol and have the potential to influence environmental process and to negatively impact human health (Després et al., 2012; Douwes et al., 2003; Fröhlich-Nowoisky et al., 2016; Shiraiwa et al., 2017). In order to understand the impact bioaerosols, such as pollen, spores, and bacteria, play on various systems, it is important to be able to identify and characterize these biological particles in the atmosphere. One common method for the detection of bioaerosols is ultraviolet laser/light-induced fluorescence (UV-LIF), because it can provide particle detection in near real-time and at high particle size resolution (Fennelly et al., 2017; Huffman and Santarpia, 2017; Sodeau and O'Connor, 2016). Many commercial UV-LIF instruments have become available for bioaerosol detection, but all of these techniques are challenged with the need to differentiate between small differences in fluorescence properties in order to sort and quantify biological aerosols from non-biological material. Recently commercialized instruments show improved ability to discriminate between particle types, for example by utilizing multiple excitation sources or other particle data (e.g. size and shape). UV-LIF techniques are inherently limited, however, by the broad nature of fluorescence spectra and so instruments face a ubiquitous problem of poor selectivity between particle types. By applying improved data thresholding and particle classification techniques, particle characterization can be further improved, but important limitations still remain (Hernandez et al., 2016; Huffman et al., 2012; Perring et al., 2015; Savage et al., 2017; Toprak and Schnaiter, 2013; Wright et al., 2014). One strategy to improving quality of differentiation between particles types has been to collect full, resolved emission spectra, each at multiple excitation wavelengths. This leads to high instrumental purchase cost, and such instruments have not been widely applied or commercialized (Huffman et al., 2016; Kiselev et al., 2013; Pan et al., 2009b; Ruske et al., 2017; Swanson and Huffman, 2018). Most commercial UV-LIF instrument for bioaerosol detection utilize 1-2 excitation wavelengths and integrate fluorescence signals into a small number of emission bands. To extend the improvements in particle classification for these commercial UV-LIF instruments, a number of multivariate analysis techniques have been applied to ambient particle analysis. The most common of these techniques include principal component analysis, factor analysis, and cluster analysis strategies. Clustering techniques, in particular, have shown successful results in providing unbiased insights to the classification of bioaerosols (Crawford et al., 2015; Pinnick et al., 2013; Robinson et al., 2013; Swanson and Huffman, 2018).

Cluster analysis is a broad class of data mining methods in which data objects placed in the same group (or cluster) are more similar to one another than to those objects placed in other groups. Clustering techniques can be divided into two central models: (1) supervised and (2) unsupervised learning. Both models have associated advantages and disadvantages. Supervised learning methods allow the "training" of data and grouping to better reflect the data observations (Eick et al., 2004; Ruske et al., 2017). This type of method enhances (trains) the clustering algorithm in that the output cluster classes are pre-determined rather than discovered, as is the case for unsupervised methods. Supervision requires the user to have appropriate starting conditions to put into the model, which are often difficult or impossible to determine. Supervised training methods are also much more time-efficient compared to unsupervised methods, which is important when analyzing ambient datasets where particle counts (individual objects) can be greater than $10^6$ (Ruske et al., 2017). In contrast, unsupervised training methods present less bias and can adapt to unique situations, because the resultant clusters are based on models that have not been previously trained. To access some of the advantages of supervised methods, however,

it is critical to first apply unsupervised models to wide collections of laboratory data of known
particle types in order to gain insight on how these models interpret data inputs and to learn how
algorithms can best be trained (Ruske et al., 2017).
Hierarchical agglomerative clustering (HAC) is an unsupervised learning method that has
been most commonly applied for bioaerosol related studies (e.g. Crawford et al., 2016; Crawford
et al., 2015; Gosselin et al., 2016; Pan et al., 2009a; Pan et al., 2007; Pinnick et al., 2013; Pinnick
et al., 2004; Robinson et al., 2013; Ruske et al., 2017). Other unsupervised clustering techniques,
such as the k-means clustering method, have shown poor results when applied to ambient data
sets because the number of clusters used to represent the data are required a priori, and this
information is usually unknown prior to analysis (Ruske et al., 2017). There are several different
HAC methods or linkages including: Single, Complete, Average, Weighted, Ward's, Centroid,
and Median (Crawford et al., 2015; Müllner, 2013). Ruske et al. (2017) compared a variety of
HAC linkages and determined that Ward's linkage had a higher percentage of correctly
classifying particles, in comparison to other HAC methods.
Recently, Savage et al. (2017) published a comprehensive laboratory study applying the
wideband integrated bioaerosol sensor (WIBS-4A) to a large and diverse set of biological and
non-biological aerosol types. Following on that work, the study presented here utilizes those data
as inputs to evaluate and challenge the HAC strategy of particle differentiation using the Ward's
linkage of unsupervised clustering. Previous HAC studies have focused primarily on (a) the
analysis of simple particle standards (i.e. fluorescent microbeads) and (b) clustering of particles
from ambient data sets. There have been relatively few published attempts to differentiate
between biological particles and interfering particles by clustering methods using controlled
laboratory UV-LIF data or to separate different kinds of biological particles from one another.
Presented here are results of the HAC method applied to data from a comprehensive WIBS
laboratory study showing that clustering can dramatically improve removal of non-biological
particle types from data sets if operated under appropriate conditions.
**2.   Experimental and Computing Methods**
The WIBS-4A (Droplet Measurement Techniques, Longmont, CO) is a commonly used UV-
LIF based instrument for the detection and characterization of biological particles. The
instrument collects particles in the size range $0.8 - 20$ μm and interrogates them in real-time as
particles flow through the path between optical sources. The WIBS collects 3 channels of
fluorescence intensity information (FL1, FL2, and FL3), particle size, and particle asymmetry for
each interrogated particle. The excitation and emission wavelengths chosen for each of the 3
fluorescence channels were designed to maximize the information gained about key biological
fluorophores present in a broad range of bioparticles (Kaye et al., 2005; Pöhlker et al., 2012). For
more information on the design, operation, and calibration of this instrument see e.g. the
manuscripts listed here and references therein (Foot et al., 2008; Healy et al., 2012a; Healy et al.,
2012b; Hernandez et al., 2016; Kaye et al., 2005; Perring et al., 2015; Robinson et al., 2017;
Savage et al., 2017; Stanley et al., 2011).
All aerosol materials utilized have been listed previously in Table 2 shown by Savage et al.
(2017), where an overview of size and fluorescence properties of particles utilized for this study
are also reported. No additional laboratory experiments were performed here beyond the results
presented previously.
The fluorescence threshold applied to the differentiation of fluorescent from non-fluorescent
particles is a key step in UV-LIF data analysis. Traditionally a fluorescence threshold has been



determined as the average baseline fluorescence intensity measured in each of three channels
during the forced trigger (FT) mode when no particles are present, plus three times the standard
deviation (σ) of that measurement (i.e. FT + 3σ) (Gabey et al., 2010). Savage et al. (2017) also
reported that additional particle discrimination is possible by using FT + 9σ as the threshold.
Both threshold definitions will be discussed here. After choosing a threshold of minimum
fluorescence, the fluorescence characteristics of a particle can be classified into 7 different
particle types introduced by Perring et al. (2015) and as summarized in Figure 1 shown by
Savage et al. (2017).
**3.  Clustering Strategy**
Hierarchical clustering methods work by grouping objects from the bottom up, meaning that
each object (particle) starts as its own "cluster," and clusters are merged together based on
similarities until a greatly reduced number of clusters are presented as a final solution. Ward's
method for clustering is among the most popular approaches for HAC and is the only method
based on a classical sum-of-squares criterion, minimizing the within-group sum of squares (or
variance) (Müllner, 2013). The WIBS-4A used here for data collection provides 5 parameters of
information for each individual particle detected (3 fluorescence channels, size and asymmetry
factor:AF), resulting in 5 dimensions of data.
The clustering analysis was performed using the open-source software R package
'fastercluster' (Müllner, 2013) using a Dell Latitude E7450 laptop computer with an Intel®
Core™ Processor (i7-5600U CPU @ 2.60 GHz, 16 GB RAM).
**3.1 Data Preparation**
Saturation of fluorescence intensity occurs at 2047 analog-to-digital counts (ADC) for each
of the three FL channels in the WIBS-4A, at which point the photomultiplier tube (PMT) reaches
its upper limit of detection. A study by Ruske et al. (2017) investigated whether non-fluorescent
(in that case, particles below the FT + 3σ fluorescence threshold) and/or saturating data points
included in the clustering analysis hindered the efficiency of the cluster output. The authors
determined that removing both saturating and non-fluorescent particles before HAC analysis
resulted in a better clustering performance in terms of correctly classifying ambient particles.
Their conclusions, however, were based on ambient field data using unknown particles types and
did not investigate laboratory-generated particles of known origin. The quality of the clustering
results are likely to be impacted by types of particles involved and the assumptions placed on
those. As shown by Savage et al. (2017), many biological particles present a large fraction that
saturate one or more of the fluorescence detectors. Conversely, many non-biological particles
present a large fraction of very weakly fluorescent particles with intensity below a given
threshold and thus that are classified as non-fluorescent. To limit pre-modification of particle
populations before clustering, the only filter applied before clustering was to remove particles
smaller than the lower particle size detection limit of the WIBS-4A (0.8 µm), similar to Ruske et
al. (2017). In contrast, both saturating and non-fluorescent particles were retained and the
clustering results will be evaluated. Figure 1 outlines the data preparation process, including the
conceptual process of normalization, clustering, and validation of data, which will be explained
in detail below.



### 3.2 Data Normalization

Normalization of the raw data is necessary before executing the clustering algorithm,
because data parameters delivered from the instrument are measured on different respective
scales. For example, fluorescent intensity values range from 0 to 2047 ADC (analog-to-digital
counts), size from 0 to ~20 µm, and AF from 0 to 100 arbitrary units. Crawford et al. (2015)
performed analysis on polystyrene latex spheres (PSLs) using several different normalization
techniques, concluding that z-score normalization is the best technique when looking at cluster
performance using Ward's linkage for the separation of PSLs. As a result, we utilize the z-score
normalization of Ward's linkage HAC for the presented study. By this type of normalization, the
mean value of all data points is subtracted from each individual data point, and then each data
point is divided by the standard deviation of all points. Standardization using the z-score method
compares results to a normal (Gaussian) population, and it therefore relies on the assumption that
input data can be described by a normal distribution (Gordon, 2006).
### 3.3 HAC Scenarios
Hierarchical agglomerative clustering performs optimally if all variables (1) are independent
of one another and (2) can be described well by a normal (Gaussian) distribution (Norusis,
2011). To achieve meaningful results from the clustering analysis data values must, therefore, be
input into the clustering algorithm with a careful understanding of how specific preparatory
conditions can significantly impact results. To investigate optimal input conditions a total of 6
clustering scenarios were explored, with conditions summarized in Table 1. The impact of two
separate variables were explored within these scenarios by varying (i) whether fluorescence
intensity were pre-normalized by particle size and (ii) whether the data values were input in
logarithmically spaced bins to produce a normal distribution.
Ambient particle distributions are well known to exhibit lognormal distributions. Further,
fluorescence intensity has been shown to scale with particle size (e.g. Hill et al., 2001;
Sivaprakasam et al., 2011). Several previous studies attempted to utilize HAC for ambient
lognormally-distributed particle size data (Crawford et al., 2014; Crawford et al., 2015; Robinson
et al., 2013), but applied the assumption that particle fluorescence is normally distributed in a
group of particles. If this assumption does not hold to be correct, however, weakly fluorescing
particles are likely to be grouped into a single cluster based on the high abundance of these
particles (Robinson et al., 2013). Scenarios C, D, and E (Table 1) utilize data input to the
clustering algorithm after fluorescence intensity was normalized to particle size in order to
explore whether the assumption that laboratory data should be treated like previously explored
ambient data sets and not logged. Scenarios B and D take into account the logging of all
parameters, producing normal distributions of all variables (AF, particle size, 3 channels of
fluorescence). For comparison, scenarios E and F explore log-spaced distributions of size and
AF, while retaining the assumption that the fluorescence output is normally distributed. Scenario
A data is neither logged nor normalized. For comparison, Scenario F represents the input
conditions that have been used frequently (e.g. Crawford et al., 2015; Ruske et al., 2017).
### 3.4 Cluster Validation
An important feature of HAC is that it provides clusters in an unsupervised manner, and the
user must determine the number of clusters that makes physical sense. One useful tool to
systematically determine the optimal number of final clusters is the Calinski-Harabasz (CH)
index, which uses the interclass-intraclass distance ratio (Liu et al., 2010). For each clustering


output the CH index was calculated for cluster solutions with one through ten clusters, and the
solution with the highest CH value was generally determined to be the optimal number of
clusters. Figure 2 shows an example CH versus cluster number plot for a mixture of *Aspergillus*
*niger* fungal spores mixed with diesel soot particles. The curve suggests the optimal result to be a
2-cluster solution for this trial, as was generally the case for investigations where two particle
types were mixed before clustering. In order to reduce the length and complexity of analysis, all
cases presented in Sections 4.1-4.3 are products of a 2-cluster solution.
**4    Results and Discussion**
The analysis of clustering quality was performed systematically and with increasing
complexity. Section 4.1 utilizes three pairs of particles types to explore the effect of particle ratio
and normalization strategies on cluster performance. Using conclusions from this section,
Section 4.2 then expands the exploration to 20 additional pairs of particle types. Section 4.3
explores the effect of three different fluorescence thresholding strategies on cluster output.
Finally, Section 4.4 investigates the ability of HAC analysis to separate particle types from
mixed populations of particle types.
**4.1 Investigating pre-normalization scenarios and particle input ratio**
To explore the ability to separate two distinct populations of particles from one another, three
different clustering trials are presented in this section as one-on-one match-ups: (1) *Aspergillus*
*niger* (fungal spores, F2) vs. NIST diesel soot (S4), (2) *Pseudomonas stutzeri* (bacteria, B3) vs.
NIST diesel soot (S4), and (3) *Aspergillus niger* (fungal spores, F2) vs. California sand (mineral
dust, D12). These four particle materials were chosen to represent key classes of coarse particles
observed in ambient air. For each trial, a given number of particles from each material type was
placed into a conceptual pool before running through the algorithm to organize clusters. The
clustering process includes: (i) evaluation of cluster performance based on particle assignment
and cluster composition, and (ii) visual representations of cluster outputs using particle type
classification introduced by Perring et al. (2015). For each of these three trials, the clustering
process was run separately using each of the six scenarios A-F described in Table 1.
Additionally, while exploring the optimal data pre-processing scenario, the influence that
different concentration ratios of particle types could play in the clustering output was also
explored. The cluster process for each trial was performed using three different ratios of particles
in each particle set including an equal ratio (50:50) and situations where the concentration of
each particle type was significantly mismatched (80:20 and 20:80). In total, this section
represents 54 individual clustering experiments (3 trials x 6 scenarios x 3 particle ratios)
exploring three independent input variables. The results will be utilized to explore many more
individual particle type match-ups in the following sections.
The first two trials include diesel soot particles, because they are commonly observed in
almost all atmospheric samples with even minimal anthropogenic influence, and because they
have fluorescence characteristics difficult to distinguish from small biological particles (e.g.
Huffman et al., 2010; Pan et al., 2012; Savage et al., 2017). For example, when excited by
photons with a wavelength of 280 nm, diesel soot can be misinterpreted as single bacterial cells
using the WIBS, and so we explored here whether the two particle types could be clustered
separately (Pöhlker et al., 2012). The three trials include two examples of biological particles,
both exhibiting fluorescent properties, but with different excitation-emission characteristics and
with different average particle size.



The output of the algorithm reports the particle type from which each particle was input in
order to evaluate the accuracy of the clustering. The resulting output of each particle with an
assigned cluster number is then compared to the originating particle type to determine
classification accuracy. Figure 3 summarizes the relative accuracy of individual clustering
experiments by representing the percent of particles misclassified with respect to known input
identities (blue bar corresponding to correct classification, red bar and overlaid value
corresponding to incorrect classification). The clustering process was generally effective for
separating particles correctly when two particle types were considered, but results vary widely
across the six scenarios. Several previous studies that used HAC to separate particles within an
ambient data set assumed that particle fluorescence is already normally distributed (Crawford et
al., 2014; Crawford et al., 2015; Robinson et al., 2013). As a result, these previous studies did
not normalize fluorescence data and thus used data preparation scenario F in their clustering
analysis. For comparison, scenarios B and D were explored to test whether the clustering
efficiency would be improved or hindered by fluorescence normalization. Scenarios A and F
produced inconsistent results, with some experiments (i.e. 50:50 ratio of fungal spores:diesel)
producing misclassification <1.1%, whereas other experiments (i.e. 20:80 ratio of
bacterial:diesel) producing misclassification >80%. In contrast, scenarios B and D produced
consistently more accurate results. Scenario B, in particular, consistently exhibited the most
accurate classification of particles for almost every individual experiment. No experiment
involving scenario B produced greater than 9% misclassification of particles, regardless of
particle input ratio, and most experiments produced results with 0.1 - 3% error. These
observations taken together suggest that particle fluorescence properties may not be well
described by normal distributions and that normalizing fluorescence data prior to analysis may
be more effective.
The results of these experiments also highlight how important the ratio of input particles can
be. While scenario B was relatively consistent, varying only between 0.1 and 3.8% error for
different ratios of the fungal spore versus diesel match-up, other experiments depended strongly
on particle ratio. It is clear that the input ratio of particle types cannot be controlled during an
ambient study, and so these results suggest that it is important to keep the possibility of varying
concentration ratios in mind when interpreting time- or air mass-associated changes in cluster
composition or when relaying the relative confidence in clustering results. For the remainder of
the discussion, experiments will be limited to a 50:50 ratio following scenario B. In each case the
number of input particles represents a random subset taken from the pool of particles in the
experimental data. As a result, individual samples selected from the same experiments (i.e. Fig.
4a, Fig 4e) can show slightly different average properties. In some cases (i.e. Diesel soot, Fig.
4d) the number of particles originally analyzed was small and so to keep the input particle ratio
50:50 the corresponding particle type was also limited to small numbers.
An important tool readily applied to analysis of ambient data is the categorization of particles
into 8 fluorescent particle types (Perring et al., 2015). Thus, to further investigate the quality of
cluster accuracy, Figure 4 shows inputs and cluster outputs from three clustering experiments
stacked as a function of fluorescence particle type and particle size. The top row of Figure 4
shows the input data for *Aspergillus niger* and diesel soot (Fig. 4a-b) paired with the outputs of
the 2-cluster solution (Fig. 4g-h). It can be seen that both particle materials have predominantly
particle type-A characteristics, meaning that they are fluorescent only in channel FL1. The
fungal material also presents roughly a third AB (green) and a small minority of non-fluorescent
(gray) characteristics. The size distribution of the fungal spores peaks at ~3 μm, whereas diesel
soot peaks at ~1 μm in size. While not shown in this plot style, the spores exhibit moderately
higher FL1 channel fluorescence, with a median of 543 ADC, whereas diesel soot exhibits a
median of 751 ADC in this channel (see Savage et al., 2017; Table 2). Both particle types show
almost no fluorescent characteristics in either FL2 or FL3. In summary, the particle distributions
are relatively similar in fluorescence particle type and their differences are largely related to
particle size, so separation of these particles through Trial 1 was hypothesized to represent a
relatively challenging initial exercise. The clustering outputs presented in Figures 4g-h, however,
visually highlight the conclusion represented by Figure 3, which is that the particles in this trial
separated very well. Cluster 1 was comprised predominantly of fungal particles and presented
fluorescence and size traits qualitatively similar to the input fungal particles, whereas cluster 2
was comprised predominantly of diesel soot particles. Results from the 50:50 ratio of the
scenario B experiments for the other two trials are also shown in the last two rows of Figure 4. In
each case, the qualitative properties of the input particles are extremely well represented by the
corresponding output cluster, corroborating the conclusion from Figure 3 that the scenario B
cases accurately separated the particle groups investigated through these experiments.
**4.2 Investigating cluster quality without fluorescence threshold**
After concluding that scenario B exhibited the most consistently accurate clustering results
using 2-cluster solutions from mixtures comprised of 2 particle type inputs, the analysis was
expanded to include a broader range of particle types. Using 50:50 ratios of two types of input
particles, prepared using scenario B (leaving fluorescence data un-normalized and forcing all
five data parameters into logarithmically spaced bins), 20 new individual experiments were
performed. The results of all 23 experiments (3 from Section 4.1 and 20 introduced in Section
4.2) are summarized in Table 2 as the percentage of particle misclassification. These trials were
chosen to represent a broad range of individual match-ups that might be expected in ambient air.
From the original 69 types of particles analyzed by Savage et al. (2017), 14 were used in
experiments here: 8 types of non-biological particles and 6 types of biological particles (2 each
of fungal spores, bacteria, and pollen species). Supplemental Figure S4 from Savage et al. (2017)
shows size distributions stacked by fluorescence particle type for each of the particle species
discussed.
Table 2a organizes clustering results into three rows, showing misclassification of F2
(*Aspergillus niger* fungal spore), B3 (*Pseudomonas stutzeri* bacteria), and P9 (*Phelum pratense*
pollen) particles, respectively, with respect to a variety of other particle types represented by
table column. Of the 15 cluster experiments between fungal spore or bacteria and non-biological
material (top two table rows), only 3 showed misclassification greater than 7.5% (bold text), and
7 were less than 3%. The three outliers were: experiment (7) F2 vs BC3 (glyoxal + ammonium
sulfate brown carbon aerosol), (8) F2 vs WT (white t-shirt particles), and (14) B3 vs WT.
Looking first at experiment (7), F2 particles show A-type fluorescence characteristics and are
dominated by a mode between 1.5 and 4 μm. BC3 particles are primarily non-fluorescent <1.5
μm, but are primarily A-type between 1.5 and 3 μm, suggesting similar size and fluorescence
properties. The white t-shirt particles separated poorly (~41% misclassification) from both the
fungal spore and bacterial particles. All three particle types (WT, F2, and B3) exhibit medium
fluorescent intensity in the FL1 channel. The poor ability to separate WT from both F2 and B3
was surprising, however, given that WT exhibited significantly higher mean fluorescence in each
of the FL2 and FL3 channels. As first mentioned by Savage et al. (2017), great care should be
taken when interpreting fluorescent particle results from indoor environments where increased



359 concentrations of bleached fibers from clothing, bedding, paper, and cleaning products may be
360 present.
361  While the results show that the spores and bacterial particles investigated could generally be
362 well separated from most potentially interfering non-biological species, the results were much
363 less successful for differentiation from pollen. P9 pollen particles separated poorly in all
364 experiments (versus D12, H2, or P5), with rate of misclassification ranging from 22 to 47%. It is
365 important to keep in mind, however, that the WIBS was operated using a standard gain setting
366 that limits analysis of particle size to below approximately 20 μm. As a result, the WIBS is
367 insensitive to whole pollen grains and so most of the particles observed during pollen
368 experiments are small pollen fragments. Any intact pollen grains that navigate the flow system to
369 be detected are likely to be binned together in the channel representing the largest particles.
370 Clustering results including pollen should be interpreted accordingly. Pollen gains can fragment
371 in ambient air as function of increased relative humidity (Miguel et al., 2006; Suphioglu et al.,
372 1992; Taylor et al., 2004), but the relative ratio of whole/fragmented particles is hard to predict
373 under ambient conditions. Smaller fragments can also exhibit different fluorescent properties
374 than whole grains (Pöhlker et al., 2013). O'Connor et al. (2014) operated a WIBS-4 (Univ.
375 Hertfordshire) at lower gain in order to improve pollen detection efficiency, but these results are
376 not explored directly here.
377  The WIBS instrument is frequently used to differentiate between airborne biological particles
378 and material of non-biological origin. A secondary goal of differentiating more finely between
379 types of biological aerosols is also frequently pursued. To investigate this goal, six additional
380 experiments were conducted by pairing two different types of non-biological particles (Table
381 2b). In contrast to the results shown in Table 2a, the clustering algorithm showed generally poor
382 ability to separate between two biological particle types. Only one of the six experiments
383 resulted in error <15% (F2 vs B3, 10.3% error), whereas error for the other five experiments
384 ranged from 18% to 65%. The worst accuracy was demonstrated by experiments (22) B1 vs B3
385 and experiment (23) P5 vs P9. Both of these experiments attempted to separate between different
386 species of a single particle type (i.e. between two bacteria or two pollen, respectively). Overall,
387 these results suggest that the clustering strategy may be quite useful at aiding the differentiation
388 of biological material from non-biological material, but that separating more finely to quantify
389 differences between types of individual biological particles is likely to be significantly more
390 challenging.

392 **4.3 Investigating impact of fluorescence thresholding strategy on cluster quality**
393  In previously published studies, removing particles from clustering analysis that exhibited
394 particle fluorescence intensity below the threshold (i.e. non-fluorescent) or at the saturating point
395 improved the efficiency of clustering (Crawford et al., 2015; Ruske et al., 2017). In Sections 4.1-
396 4.2, particles with either of these characteristics were left in the analysis to prevent the
397 underestimation of particles clustered. In this section, however, we investigated whether
398 removing non-fluorescent particles could improve cluster accuracy for the experiments that
399 performed poorly in Section 4.2. Of the 23 trials represented in Table 2, 10 experiments
400 exhibited 15% or greater misclassification and were subjected to further analysis in order to
401 investigate whether using a more discriminating fluorescence thresholding strategy could
402 improve cluster results. In all 10 cases fluorescence saturating particles were retained, and three
403 separate thresholding conditions were compared by: (I) keeping all non-fluorescent and
404 saturating particles, (II) removing non-fluorescent particles by applying a fluorescence threshold



of FT baseline + 3σ, and (III) and removing non-fluorescent particles by applying a fluorescence
threshold of FT baseline + 9σ. Table 3 shows the percentage of particles misclassified in each of
three scenarios (Table 3a) as well as the number of particles subjected to the clustering algorithm
(Table 3b).
Each scenario, with exception of the B3 vs B9 experiment (21), shows a decrease in particle
misclassification from scenario I (no fluorescence threshold applied) to scenario II (FT + 3σ). In
contrast, eight of the ten scenarios *increase* in particle misclassification when raising the
fluorescence threshold from 3σ (II) to 9σ (III). The exceptions to this trend are experiments (8)
F2 vs WT and (19) F2 vs P9, which show nominal improvement in error (2-4% reduction) with
increased threshold. We hypothesize that the 9σ results degrade, in most cases, because the
threshold becomes high enough that most weakly fluorescing particles have been removed from
analysis. This reduces the ability of the cluster to group into low and high fluorescence
categories, and so remaining particles are separated less efficiently. Secondly, removing particles
at higher fluorescence thresholds leads to increasingly poor counting statistics, as represented in
Table 3b by the number of particles included in each experiment. Overall, these results suggest
that inputting particles into the clustering analysis with at least a nominal fluorescence threshold
(i.e. FT + 3σ) can improve the clustering results in many cases, however, increasing the
threshold further may decrease cluster quality.
**4.4 Investigating cluster ability to separate complex synthetic mixtures**
To this point, our investigation has focused on a variety of individual match-ups between two
distinct particle types. To better simulate real-world scenarios, we analytically synthesized six
mixtures of particles by pooling existing data from selected particle types in prescribed ratios.
Each mixture was synthesized to roughly represent a different hypothetical mixture of particles
that might be expected. Table 4 provides an overview of the percentage of each particle type
included as well as the total number of particles in the mixture. Mixtures 1 and 2 were
synthesized arbitrarily to test if a minority (25%) of one type of fungal spores (F2) could be
separated from a majority (75%) of a mixture of three different non-biological materials.
Mixtures 3 and 4 synthesized arbitrary mixtures of two types of bioaerosol (F2 and B3) with
three or five types of non-biological particles, respectively. Mixture 5 was synthesized to
examine the separation of pollen (P9) from a set of five non-biological particles. Mixture 6 was
synthesized to simulate an indoor environment that might have a mixture of biological particles
(F2 and B3) with non-biological materials, including bleached fibers (WT). These mixtures are
not intended to closely mimic any set of individual ambient conditions, but are rather used as
very rough synthetic scenarios used for discussion. In a real-world sampling environment one
would also expect a high concentration of non-fluorescent particles as well (e.g. most organic
aerosols, sea salt, dusts), but these were largely not sampled as a part of the Savage et al. (2017)
study, which focused on fluorescent particles. As a result, relatively non-fluorescent particles
like D12 and H2 were included here as "fillers" in most mixtures as surrogates for other types of
non-fluorescent particles. Clustering analysis was performed using the ratios listed in Table 4,
the B scenario of pre-normalization conditions, and filtering non-fluorescent particles below the
FT + 3σ threshold. In all cases, the number of clusters retrieved after HAC was the same as the
number of particle types input.
Cluster results from all six mixtures are summarized in Figure 5. Figure 5 (Part A) shows the
number of particles from each type assigned to each cluster, and Parts B and C show results
grouped by general particle classification (brown for non-biological and dark green for



biological). Overall, the ability of the HAC analysis to separate the biological particles from the
non-biological particles was high. In some cases the quality of separation of one or two
biological species from a mixture of non-biological materials was even higher than the 2-
material match-ups shown in Sections 4.1-4.3. The two 4-component mixtures showed 22.4%
and 14.8% misclassification of fungal spores. In both cases, a small fraction of each of the non-
biological materials were mixed into the spore cluster, whereas almost none (1.5% and 0.6%) of
the spores were incorrectly mixed into the sum of the non-biological clusters.
Mixtures 3 and 4 showed similar misclassification for fungal spores (11.9% and 13.8%,
respectively), whereas the bacterial particles clustered with amazing quality. For Mixture 3, no
particles other than bacterial particles were grouped into Cluster 1, and only 16 of 213 bacterial
particles were assigned to other clusters. For Mixture 4, 135 of 137 particles in Cluster 6 were
bacterial in origin and 135 of 142 bacterial particles were assigned to the cluster. The
combination of fungal and bacterial particles in Mixtures 3 and 4 resulted in a total of 5.0% and
5.3% misclassification of all biological particles.
In contrast to the poor separation of pollen from other particle types discussed in Section 4.2,
Mixture 5 showed a higher quality of separation between pollen (9.4% misclassified) and the
sum of five other non-biological particle types. Lastly, the mixture designed to roughly mimic an
indoor environment including white t-shirt particles. In this mixture the WT particles confounded
the spore separation, but the bacterial separation was nearly flawless.
Another surprising observation from the analysis of these synthetic mixtures was that the
diesel soot particles (Mixtures 1, 2, 4, and 5) separated into their own cluster in almost all cases
with very high quality (1.8%, 2.9%, 0.6%, and 9.4%, respectively, of diesel soot particles
misclassified into a different cluster). The quality of separation of bacterial particles and diesel
soot (Mixture 4) was especially amazing, given the qualitative similarity of the two particle
populations. For example, size-distributions of each particle type show primarily A-type particles
with similar mean fluorescent intensity values in FL1, FL2, and FL3 (Savage et al., 2017).

## 5. Conclusions

Application of results from a recent set of systematic laboratory experiments (Savage et al.,
2017) by the commonly used hierarchical agglomerative clustering analysis helps to reveal areas
where the tool can be used well and other areas where it struggles. First (Section 4.1) it was
observed that differing ratios of particle input into the clustering algorithm can produce
dramatically different results. It will be important for anyone applying HAC to ambient particle
sets where particle ratios are not independently verified to interpret results somewhat loosely. In
Section 4.1 the clustering quality of scenario B, where fluorescence intensity was not normalized
to particle size and where all input variables were binned into log space, was determined to
consistently demonstrate the highest quality results. Further, the ability to the HAC analysis to
separate between two groups of individual particle types using no fluorescence threshold
(Section 4.2) and comparing three separate threshold strategies (Section 4.3) was shown to be
relatively high in many cases, but confounded in others. Lastly, Section 4.4 explored the ability
of HAC analysis to separate biological components from more complex mixtures of four to
seven types of input particles.
A standard fluorescence threshold of FT + 3σ has been commonly applied during WIBS
analysis to separate between fluorescent and non-fluorescent particles. Savage et al. (2017)
concluded that application of a more aggressive threshold strategy (FT + 9σ) could help
discriminate between biological and non-biological particles more successfully in many



circumstances, however certain types of interfering, non-biological particle species can still
confound WIBS analysis irrespective of the threshold. Here we have investigated an orthogonal
strategy to separate particle types by subjecting particles to HAC computer analysis. By
comparing the results of the HAC analysis with raw separation based on fluorescence
thresholding alone, the HAC analysis can clearly increase quality of differentiation. Interestingly,
while Savage et al. (2017) reported that the FT + 9σ strategy helped improved differentiation,
using the same threshold in conjunction with HAC analysis actually degraded results. We
therefore conclude that if HAC analysis is to be performed, the standard FT + 3σ threshold is
likely to produce the highest quality results, however if HAC is not to be applied that the FT +
9σ threshold is the most likely to reduce a large fraction of non-biological particles.
The overall message here is that HAC can be applied successfully to differentiate particle
types sampled by WIBS instruments and that it is most successful at separating biological
species (i.e. fungal spores and bacteria) from non-biological particles. In all cases the HAC
method allows separation of particles at least at the order-of-magnitude level, and often with
misclassification of <5%. As mentioned by Savage et al. (2017), however, it should always been
kept in mind that different instruments may produce slightly different signals due to physical
differences (i.e. fluorescence calibration, tuning, and detector gain sensitivity) and that results
here are only generally extendable to other UV-LIF instruments(Robinson et al., 2017). Subtle
differences in particles observed in a real-world environment may complicate HAC analysis or
the extension of results presented here. The UV-LIF community is encouraged to continue
laboratory investigations, including detailed interrogation of clustering analytical techniques, to
further understand limitations to better differentiating between particles.
**6. Acknowledgments**

The authors acknowledge the University of Denver for financial support from the faculty
start-up fund. Nicole Savage acknowledges financial support from the Phillipson Graduate
Fellowship at the University of Denver. Martin Gallagher and David Topping in the School of
Earth and Environmental Sciences at the University of Manchester are acknowledged for initial
discussion regarding clustering strategy. Cathy Durso at the University of Denver Center for
Statistics and Visualization is acknowledged for help running clustering algorithms. All
contributors to the Savage et al. (2017) paper, in which all experimental data discussed here were
originally presented, are acknowledged for their contributions.





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




**Tables**
Table 1. Six scenarios explored, with varying combinations of pre-analysis treatment. (1)
Fluorescence normalization refers to whether fluorescence intensity was normalized to particle
size. (2) Variables logged refers to whether data was manipulated to produce a normal
distribution.

| Parameters | A | B | C | D | E | F |
|---|---|---|---|---|---|---|
| 1. Fluorescence Normalization<br>2. Variables Logged | 1. No<br><br>2. No | 1. No<br><br>2. Yes | 1. Yes<br><br>2. No | 1. Yes<br><br>2. Yes | 1. Yes<br><br>2. Yes, only AF/Size variables | 1. No<br><br>2. Yes, only AF/Size variables |




Table 2. Misclassification of 2-cluster solutions for 23 match-ups of two individual particle types
(equal ratio of particle number, B-scenario). Misclassification calculated as the sum percentage
of particles misclassified in each cluster divided by the total number of particles. Three
biological particle types (F2, B3, P9) compared separately to (a) non-biological particle materials
and (b) biological particle materials. Particle number input was a subset of total population of
particles experimentally analyzed.

(a)

| | Non-biological particle materials | | | | | | | |
|---|---|---|---|---|---|---|---|---|
| | Diesel soot (Soot 4) S4 | California sand (Dust 2) D2 | Arizona Test Dust (Dust 12) D12 | Suwannee River Humic Acid (HULIS 2) H2 | Methyl-glyoxal + glycine aerosol (Brown carbon 1) BC1 | Glyoxal + amm. sulfate aerosol (Brown carbon 3) BC3 | White t-shirt (Misc. 2) WT | Wood smoke (Soot 6) WS |
| *Aspergillus niger* (Fungi 2) | *(1)* *0.1%* | *(3)* *2.6%* | (4) 6.1% | (5) 4.8% | (6) 2.5% | (7) **23.0%** | (8) **40.5%** | (9) 7.2% |
| *P. stutzeri* (Bacteria 3) | *(2)* *1.2%* | | (10) 1.9% | (11) 1.2% | (12) 1.3% | (13) 6.1% | (14) **41.7%** | (15) 4.7% |
| *Phelum pretense* (Pollen 9) | | | (16) **22.7%** | (17) **23.2%** | | | | |

(b)

| | Biological particle materials | | | | |
|---|---|---|---|---|---|
| | *S. cerevisiae* (Fungi 4) F4 | *Phelum pretense* (Pollen 9) P9 | *P. stutzeri* (Bacteria 3) B3 | *Taxus baccata* (Pollen 5) P5 | *B. atrophaeus* (Bacteria 1) B1 |
| *Aspergillus niger* (Fungi 2) | (18) **27.9%** | (19) **36.4%** | (20) 10.3% | | |
| *P. stutzeri* (Bacteria 3) | | (21) **18.3%** | | | (22) **65.4%** |
| *Phelum pratense* (Pollen 9) | | | | (23) **46.8%** | |






Table 3. Further exploration of 2-cluster solutions for the 10 match-ups of two individual particle types shown in Table 2 with misclassification >15%. Each match-up shown using three separate fluorescence threshold strategies in advance of particle input into cluster algorithm: (I) all particles included (no fluorescence threshold), (II) particles with fluorescence intensity < FT + 3σ removed, and (III) particles with fluorescence intensity < FT + 9σ removed. (a) Particle misclassification. (b) Total particle number used for clustering experiment.

(a) Percent misclassified

Bio + Non-bio

| Input | (7) F2 + BC3 | (8) F2 + WT | (14) B3 + WT | (16) P9 + D12 | (17) P9 + H2 |
|---|---|---|---|---|---|
| (I) All particles | 23.0% | 40.5% | 41.7% | 22.7% | 23.2% |
| (II) Fluor. > FT + 3σ | 10.3% | 36.2% | 24.3% | 19.3% | 3.4% |
| (III) Fluor. > FT + 9σ | 41.4% | 32.6% | 31.8% | 45.3% | 14.0% |

Bio + Bio

| Input | (18) F2 + F4 | (19) F2 + P9 | (21) B3 + P9 | (22) B1 + B3 | (23) P9 + P5 |
|---|---|---|---|---|---|
| (I) All particles | 27.9% | 36.4% | 18.8% | 65.4% | 46.8% |
| (II) Fluor. > FT + 3σ | 13.3% | 31.0% | 20.0% | 77.5% | 24.9% |
| (III) Fluor. > FT + 9σ | 29.0% | 28.6% | 29.0% | 66.7% | 33.9% |

(b) Number of particles

Bio + Non-bio

| Input | (7) F2 + BC3 | (8) F2 + WT | (14) B3 + WT | (16) P9 + D12 | (17) P9 + H2 |
|---|---|---|---|---|---|
| (I) All particles | 1,959 | 565 | 565 | 10,359 | 8,902 |
| (II) Fluor. > FT + 3σ | 1,000 | 393 | 393 | 171 | 207 |
| (III) Fluor. > FT + 9σ | 471 | 319 | 319 | 38 | 37 |

Bio + Bio

| Input | (18) F2 + F4 | (19) F2 + P9 | (21) B3 + P9 | (22) B1 + B3 | (23) P9 + P5 |
|---|---|---|---|---|---|
| (I) All particles | 10,000 | 8,900 | 10,000 | 10,000 | 10,000 |
| (II) Fluor. > FT + 3σ | 9,600 | 8,500 | 9,800 | 10,000 | 10,000 |
| (III) Fluor. > FT + 9σ | 9,200 | 8,100 | 9,700 | 10,000 | 7,895 |



Table 4. Particle fraction for each type and total particle number used as inputs for synthetic
mixtures.

| Mixture Number | Mixture Name | F2 *Asp. niger* (Fungi) | B3 *P. stutzeri* (Bacteria) | P9 *Phelum pretense* (Pollen) | S4 Diesel soot | D12 AZ Test Dust | H2 Suwannee River Humic Acid | BC1 Brown Carbon 1 | WS Wood smoke | WT White t-shirt | Total Particle Number |
|---|---|---|---|---|---|---|---|---|---|---|---|
| 1 | 4-Comp. A | 25% | | | 25% | 25% | 25% | | | | 680 |
| 2 | 4-Comp. B | 25% | | | 25% | 25% | | | 25% | | 680 |
| 3 | High PBAP | 25% | 25% | | | 20% | 20% | 10% | | | 850 |
| 4 | Low PBAP | 12.5% | 12.5% | | 15% | 15% | 15% | 15% | 15% | | 1134 |
| 5 | Pollen | | | 30% | 10% | 20% | 20% | 10% | 10% | | 850 |
| 6 | Indoor Air | 20% | 20% | | | 20% | 20% | | | 20% | 850 |






**Figures**

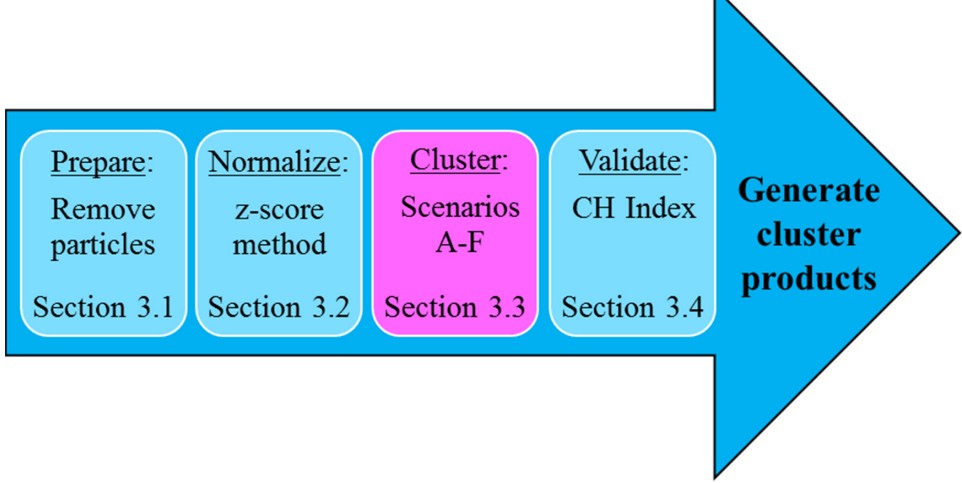

Figure 1. Schematic diagram showing the data preparation process resulting in the generated
clustering products. Parameters within the pink box are the focus of this manuscript.





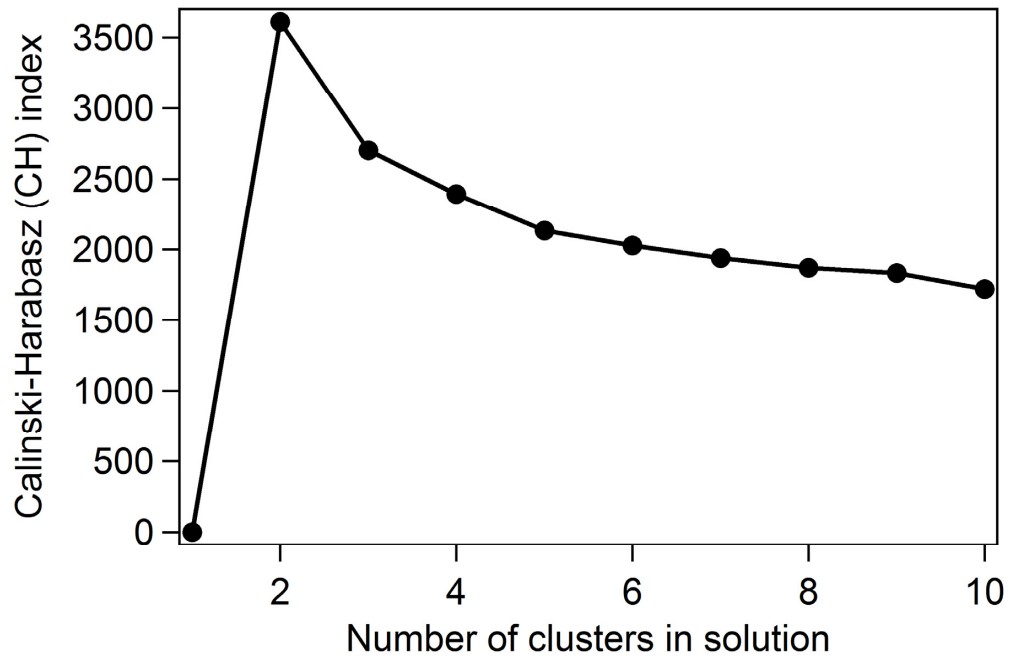


Figure 2. Example of Calinski-Harabasz Index plot for cluster experiment with input of
*Aspergillus niger* and diesel soot (50:50 ratio). Optimal number of clusters is determined by the
highest CH value.

|  | A | B | C | D | E | F |
|---|---|---|---|---|---|---|
| **Fungi : Diesel** |  |  |  |  |  |  |
| 50:50 Ratio | 1.1 | 0.9 | 7.2 | 4.5 | 3.6 | 0.8 |
| 80:20 Ratio | 64.8 | 4.1 | 4.5 | 2.9 | 3.8 | 76.5 |
| 20:80 Ratio | 2.1 | 3.8 | 68.5 | 6.0 | 19.5 | 2.1 |
| **Bacteria : Diesel** |  |  |  |  |  |  |
| 50:50 Ratio | 50.0 | 1.2 | 6.8 | 4.5 | 31.6 | 50.0 |
| 80:20 Ratio | 0.2 | 0.2 | 0.7 | 1.0 | 0.9 | 0.2 |
| 20:80 Ratio | 80.0 | 0.3 | 68.2 | 0.3 | 43.7 | 80.0 |
| **Fungi : Dust** |  |  |  |  |  |  |
| 50:50 Ratio | 12.7 | 2.6 | 24.3 | 23.5 | 18.4 | 30.6 |
| 80:20 Ratio | 76.6 | 9.0 | 20.0 | 25.4 | 25.4 | 29.3 |
| 20:80 Ratio | 35.9 | 1.5 | 55.7 | 23.4 | 44.6 | 58.6 |


Figure 3. Cluster misclassification shown for three combinations of fungal spores (F2), bacteria
(B3), and diesel soot (S4). Each combination explored with respect to ratio of input particle
number using the scenario B and a 2-cluster solution for each experiment. Scenario letter A-F
refers to scenarios summarized in Table 1. Red shaded region (and values) indicates the percent
of particles misclassified. Blue shaded region represents the percentage of particles correctly
classified.





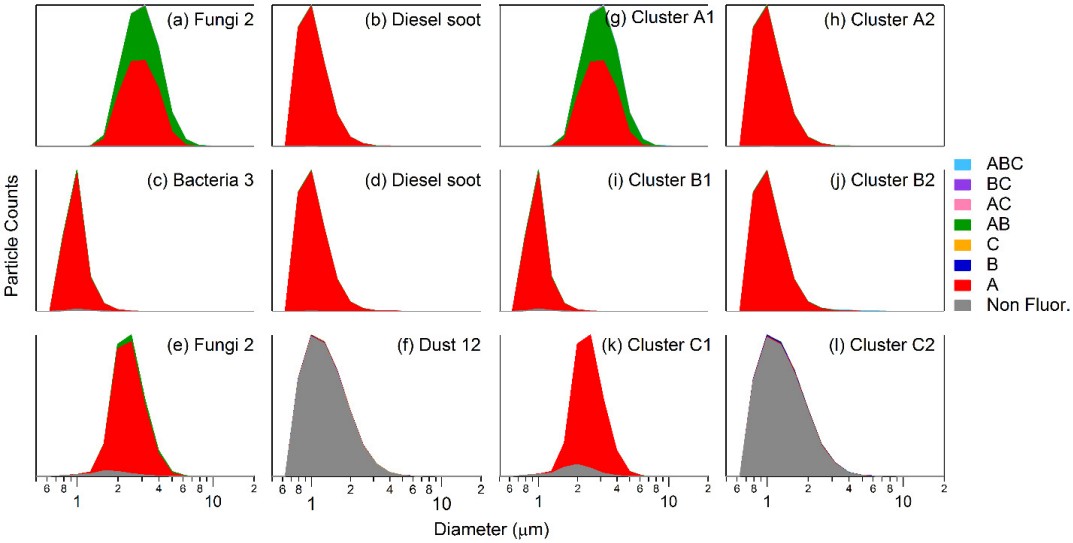


Figure 4. Particle type stacked category size distributions for input and output clustering results,
using FT + 3σ threshold definition. Each experiment (row) shows match-ups of two particle
types using 50:50 ratios, scenario B, and 2 cluster solutions. Left two columns show properties of
input particles, right two columns show properties of cluster outputs.



**Part A: Individual Clusters** (Particle Number)  
**Part B: Grouped Clusters** (Particle Number)  
**Part C: Summary** (Cluster Quality)

**Mixture #1: 4 Comp. - A**

| Cluster | F2 | S4 | D12 | H2 |
|---|---|---|---|---|
| 1 | 163 | 2 | 22 | 23 |
| 2 | 7 | 1 | 123 | 67 |
| 3 | 0 | 0 | 21 | 80 |
| 4 | 0 | 167 | 4 | 0 |

| Cluster | Fungi | | | Non-bio |
|---|---|---|---|---|
| 1 | 163 | | | 47 |
| 2-4 | 7 | | | 463 |

**Mixture #1**

| | Total P. | Miscl. | Cat. |
|---|---|---|---|
| | 210 | 22.4% | Fungi |
| | 470 | 1.5% | Non-bio |

**Mixture #2: 4 Comp. - B**

| Cluster | F2 | S4 | D12 | WS |
|---|---|---|---|---|
| 1 | 167 | 2 | 23 | 4 |
| 2 | 2 | 3 | 88 | 10 |
| 3 | 1 | 0 | 55 | 156 |
| 4 | 0 | 165 | 4 | 0 |

| Cluster | Fungi | | | Non-bio |
|---|---|---|---|---|
| 1 | 167 | | | 29 |
| 2-4 | 3 | | | 481 |

**Mixture #2**

| | Total P. | Miscl. | Cat. |
|---|---|---|---|
| | 196 | 14.8% | Fungi |
| | 484 | 0.6% | Non-bio |

**Mixture #3: High PBAP**

| Cluster | F2 | B3 | D12 | H2 | BC1 |
|---|---|---|---|---|---|
| 1 | 0 | 197 | 0 | 0 | 0 |
| 3 | 200 | 6 | 13 | 2 | 6 |
| 2 | 9 | 10 | 133 | 79 | 6 |
| 4 | 4 | 0 | 21 | 88 | 25 |
| 5 | 0 | 0 | 3 | 1 | 47 |

| Cluster | Fungi | Bacteria | | Bio | Non-bio |
|---|---|---|---|---|---|
| 1 | 0 | 197 | | | 0 |
| 3 | 200 | 6 | | | 21 |
| 2,4,5 | 13 | 10 | | | 403 |
| 1,3 | | | | 403 | 21 |

**Mixture #3**

| | Total P. | Miscl. | Cat. |
|---|---|---|---|
| | 227 | 11.9% | Fungi |
| | 197 | 0.0% | Bacteria |
| | 424 | 5.0% | Bio |
| | 426 | 5.4% | Non-bio |

**Mixture #4: Low PBAP**

| Cluster | F2 | B3 | S4 | D12 | H2 | BC1 | WS |
|---|---|---|---|---|---|---|---|
| 1 | 0 | 0 | 0 | 10 | 15 | 20 | 0 |
| 2 | 23 | 2 | 0 | 125 | 77 | 6 | 165 |
| 3 | 0 | 0 | 0 | 3 | 1 | 128 | 1 |
| 4 | 4 | 0 | 0 | 18 | 68 | 11 | 2 |
| 5 | 3 | 0 | 169 | 8 | 9 | 0 | 0 |
| 6 | 0 | 135 | 1 | 0 | 0 | 0 | 1 |
| 7 | 112 | 5 | 0 | 6 | 0 | 6 | 1 |

| Cluster | Fungi | Bacteria | | Bio | Non-bio |
|---|---|---|---|---|---|
| 7 | 112 | 5 | | | 13 |
| 6 | 0 | 135 | | | 1 |
| 1-5 | 30 | 2 | | | 836 |
| 6,7 | | | | 252 | 14 |

**Mixture #4**

| | Total P. | Miscl. | Cat. |
|---|---|---|---|
| | 130 | 13.8% | Fungi |
| | 136 | 0.7% | Bacteria |
| | 266 | 5.3% | Bio |
| | 868 | 3.7% | Non-bio |

**Mixture #5: Pollen**

| Cluster | P9 | S4 | D12 | H2 | BC1 | WS |
|---|---|---|---|---|---|---|
| 1 | 0 | 0 | 13 | 16 | 13 | 0 |
| 2 | 2 | 0 | 28 | 83 | 15 | 1 |
| 3 | 0 | 0 | 4 | 1 | 51 | 1 |
| 4 | 6 | 2 | 113 | 70 | 0 | 79 |
| 6 | 5 | 77 | 3 | 0 | 0 | 0 |
| 5 | 242 | 6 | 9 | 0 | 6 | 4 |

| Cluster | | | Pollen | | Non-bio |
|---|---|---|---|---|---|
| 5 | | | 242 | | 25 |
| 1-4,6 | | | 13 | | 570 |

**Mixture #5**

| | Total P. | Miscl. | Cat. |
|---|---|---|---|
| | 267 | 9.4% | Pollen |
| | 583 | 2.2% | Non-bio |

**Mixture #6: Indoor Air**

| Cluster | F2 | B3 | D12 | H2 | WT |
|---|---|---|---|---|---|
| 1 | 160 | 7 | 13 | 0 | 31 |
| 4 | 0 | 154 | 0 | 0 | 0 |
| 2 | 4 | 0 | 32 | 95 | 35 |
| 3 | 6 | 9 | 125 | 75 | 62 |
| 5 | 0 | 0 | 0 | 0 | 42 |

| Cluster | Fungi | Bacteria | | Bio | Non-bio |
|---|---|---|---|---|---|
| 1 | 160 | 7 | | | 44 |
| 4 | 0 | 154 | | | 0 |
| 2,3,5 | 10 | 9 | | | 466 |
| 1,4 | | | | 321 | 44 |

**Mixture #6**

| | Total P. | Miscl. | Cat. |
|---|---|---|---|
| | 211 | 24.2% | Fungi |
| | 154 | 0.0% | Bacteria |
| | 365 | 12.1% | Bio |
| | 485 | 3.9% | Non-bio |

Figure 5. Overview of synthetic mixtures. Six mixtures shown as groups of rows, with input particle fractions defined in Table 4. Part A (left columns) show particle number retrieved by each individual cluster and categorized by each input particle type. Part B (middle columns) show particle number categorized and grouped by particle classes (i.e. non-biological and biological). Part C (right columns) show misclassification of groups of particles. Colors: light green (fungal spores), blue (bacteria), pink (pollen), dark green (grouped biological), brown (all non-biological).