# Peer review of "Title: Evaluation of a Hierarchical Agglomerative Clustering Method Applied to WIBS"

_Atmospheric Measurement Techniques, 2018_

## Referee Comment (RC1) · Anonymous Referee #3 · 9 May 2018

This paper describes methods and results which should help improve the interpretation and use of data obtained with UV-LIF instruments such as the WIBS. The WIBS measures light scattering, a light-scattering asymmetry factor, and fluorescence in three channels. Fielded instruments with data rates that can exceed hundreds of particles per minute are available. This paper uses a large set of WIBS data measured for individual materials (Savage et al. 2017) to evaluate different preprocessing procedures for analysis of such data. Mathematical simulations of externally mixed particles of

known composition are studied. The findings should be useful not only for understanding WIBS data, but more broadly in applying Hierarchical Agglomerative Clustering to some other problems in aerosol analytical chemistry. I recommend publication. However, I request that several confusing items be made less confusing.

The use of the term "synthetic mixtures" (L31-32, L424, 707, L734) is confusing. Chamber studies with synthetic mixtures of real aerosols and real gases are not uncommon in aerosol science. A google search of "synthetic mixture" provides discussions of various real "synthetic mixtures." I only looked at the first 8 or so items in that search, but I saw none with the meaning used in this paper. The online dictionaries I saw do not indicate this use of "synthetic" (which as far as I can tell indicates something about numerical or computational). Synthetic organic chemists make real chemicals. If "synthetic mixtures" is used for the simulated data investigated here, what terminology is left for researchers to use when they make real synthetic mixtures of aerosols in a chamber and investigate changes in clusters as time passes and as particles agglomerate? I do not see how a reader can see from the abstract or even well into this paper that "synthetic" is being used in this highly non-standard way, and that Savage et al., 2017 did not measure mixtures of particles. The "synthetic mixtures" are actually numerical (or mathematical) simulations of the WIBS the data that should be obtained for dilute mixtures of particles. Real mixtures of particles can form agglomerates, and some may agglomerate quickly unless they are sufficiently dilute.

L 20-22 (Abstract). "Here we show for the first time a systematic application of HAC to a comprehensive set of laboratory data collected using the wideband integrated bioaerosol sensor (WIBS-4A) (Savage et al., 2017)." Suggest change to: "Here we show for the first time a systematic application of HAC to a comprehensive set of laboratory data collected for individual particle types using the wideband integrated bioaerosol sensor (WIBS-4A) (Savage et al., 2017). Here the WIBS data for single-composition aerosols is combined numerically to generate data to simulate WIBS values for mixtures of aerosol."

L31-32 (Abstract): "Lastly, six synthetic mixtures of four to seven components were analyzed." Might be changed to: "Numerical simulations of mixtures of four to seven components were HAC analyzed."

L424: "Investigating cluster ability to separate complex synthetic mixtures" Might be changed to: Investigating the capability to separate particles in simulations of complex synthetic mixtures

L426-429: "To better simulate real-world scenarios, we analytically synthesized six mixtures of particles by pooling existing data from selected particle types in prescribed ratios. Each mixture was synthesized to roughly represent a different hypothetical mixture of particles that might be expected." "Analytically" suggests equations or functions were used in obtaining the data for the mixtures. Isn't "numerically" or "computationally" what is meant?

L426-429 might be changed to: "To better simulate real-world scenarios, we numerically simulated six mixtures of particles by pooling existing WIBS data from selected particle types in prescribed ratios. Each simulated mixture was assembled to roughly represent a different hypothetical mixture of particles that might be expected. Also, the particles in each simulated mixture are assumed to be so dilute that any agglomeration is negligible." Also, a significant fraction of readers read the abstract and then look at the figures to see what the results will be. Adding clarifying words to the figure captions and tables would be useful.

I don't know what "normalized to particle size" means here. Please clarify, possibly with an equation. Please also give the ranges of error in particle sizes expected. Why is scenario D worse than B? I think it is because D adds noise to the FL signals, making them less informative by decreasing the S/N. This added noise occurs in the elastic scattering measurements, and also results from the approximations used in estimating solutions to the inverse problem for size (with unknown shape, orientation and refractive index). If the scattering measurement and the solution to the inverse problem were

perfect, then D and B should give very similar results, at least for spherical particles and some methods of normalizing to particle size and shape. It may be useful to cite a paper or data with WIBS measurements of size and fluorescence for uniformly-sized fluorescent PSL. For a single size of PSL, do plots of the WIBS-measured scattering and fluorescence fall on a line or are they spread more randomly? Even for a spherical PSL particle, with known refractive index, would you suspect that the noise is large enough to make D less useful than B?

Can the authors say anything about the length of times bacteria or fungal spores might last in an urban environment before a significant fraction of the bioparticles combine with soot, and how that might affect the usefulness of the WIBS? I'll be very interested to see the results when (sometime in the future) the authors inject bacteria or fungal spores into a chamber, add soot particles, use the WIBS to sample with time, and then repeat the some of the analyses in this paper with the results given as a function of time.

L23: In abstract: "ratio" of what? In the text, "ratio" first appears in "distance ratio." Suggest change first use of "ratio" in abstract to "ratio of particle concentrations."

L117: please add wavelength ranges of FL1 to FL3. Aim for a little broader set of readers.

L171: replace "will be" with "were".

L199: Suggest change to: Ambient particle number vs size distributions can often be well approximated by lognormal distributions (citation), although specific subsets of particles, such as bacteria, pollens or fungal spores, may not exhibit lognormal distributions.

L245: "placed into a conceptual pool"? How about, "A subset of the particles were selected randomly for analysis"?

L258-259: "diesel soot particles . . . commonly observed . . ." Is this referring to WIBS

measurements? Please provide a citation(s).

L299-300: Do you mean: "In each case the input particles are a random subset . . ."

Please also note the supplement to this comment:
https://www.atmos-meas-tech-discuss.net/amt-2018-109/amt-2018-109-RC1-supplement.pdf

---

## Referee Comment (RC2) · Anonymous Referee #1 · 1 Jun 2018

This paper builds on existing literature examining unsupervised learning techniques to improve the interpretation and classification of data obtained with WIBS UV-LIF spectrometers. As shown in previous publications, Hierarchical Agglomerative Clustering (HAC) can serve as a robust data analysis method for classification/interpretation of bioaerosol data but the accuracy of technique is highly sensitive to the choice of clustering linkage and data pre-treatment (e.g., Crawford et al., 2015); this is further explored in this paper which elucidates how data pre-treatment choices such as choice

of fluorescent threshold and log normalising data may influence clustering accuracy using laboratory samples of known particle types (Savage et al., 2017) in various synthetic mixtures, and thus the authors present tentative recommendations of data pretreatment regimes depending on the analysis goals. Overall the paper is well written and the computational experiments well thought out. The findings here are useful and further validate the usefulness of Hierarchical Agglomerative Clustering for interpretation of WIBS data. The results also provide a useful framework for testing Hierarchical Agglomerative Clustering data pre-treatment regimes for other atmospheric science data problems and neatly demonstrate the potential pitfalls of not rigorously performing such tests. I recommend publication after the following comments have been addressed.

Specific comments

L73-77: The authors have conflated some of the terminology relating to unsupervised and supervised leaning methods. I'm uncomfortable with the use of the term clustering when discussing supervised methods as clustering specifically relates to cluster analysis. I suggest replacing "clustering techniques" with "classification algorithms" and "(trains) the clustering algorithm" with "(trains) the classification algorithm".

L120: Please state the bands and what they relate to.

L198: Can the authors please clarify why they have used log spaced bins. Do you mean that you have taken a log of the data and it is binned naturally by the discrete nature of the detector resolution (i.e., fine bins) or have you binned the data into specific (coarse) log bins? If it is the latter can you please state what the bins are and can you comment on how forcing the data to in bins may influence the clustering? My concern here is that too coarsely binning the data may create artificial hotspots due to reduced resolution and bias the clustering, reducing the capacity to differentiate between particles with similar properties. Can the authors comment on this and demonstrate the effect this may have by providing an example for comparison where the data is converted to log space and not binned. I also wonder if the bins should be normalised by the bin width to further complicate matters.

L254: Can the authors comment on the environmental applicability of the chosen ratios. I would suspect that in an urban environment you may expect something closer to a ratio of 1:99 fungal to diesel particles with the converse being true in a forest environment. How does the clustering perform under such extreme mismatches?

L238: Would it be possible to show examples of the cluster centroids for a case where there is significant misclassification? This may illuminate why the algorithm is failing to correctly attribute particles. It may also be useful to examine the fluorescence/AF characteristics of each cluster as a function of size here. A 2D histogram or color density plot could show distinct hot spots that haven't been separated correctly and could provide a basis for manual separation based on sensible thresholds.

L312-315: Can you describe the method for producing the soot as they seem rather large as compared to that in the study of Toprak and Schnaiter (2013) which were also coincidently found to be weakly fluorescent in FL1. Perhaps the soot used in this study is larger and more fluorescent than we may expect of ambient/urban soot which may cause some of the difficulty in correctly attributing in in some cases?

L384: Would we expect to be able to differentiate between 2 different particles of the same type with such coarse spectral resolution?

L415: Again I wonder if the use of too coarsely separated bins may compromise the 9-sigma thresholding and cause misclassification?

L514: Can the authors comment on the applicability of their findings to new high resolution UV-LIF instruments that are beginning to become commercially available. Some of these new instruments have significantly more channels/greater fluorescent resolution than the WIBS.

Technical corrections

L63: instruments, not instrument.

L370: grains, not gains.

L112: Suggest "Experimental and Computational Methods"

L131: "each of the three"

L181: "was the best"

---

## Referee Comment (RC3) · Anonymous Referee #2 · 3 Jun 2018

This manuscript discusses application of Hierarchical Agglomerative Clustering (HAC) to analysis of data collected using the Wideband Integrated Bioaerosol Sensor (WIBS-4A). While real-time detection of bioaerosols has been quite well controlled, the analysis and classification is still challenging and vital problem. Therefore, investigation and improvements in this area are very important and crucial for understanding the abilities and limitations of LIF aerosol detectors. The manuscript is well written and in detail reveals important problems of fluorescence data analysis of bioaerosols. I rec-

ommend presented manuscript to publication, however some corrections and further explanations to the following remarks will be appreciated:

1. The techniques of single particle detection using LIF devices, like WIBS, reached relatively high reliability and perfection. The device collects data in real time, on the other hand the presented results are offline. The data analysis takes a long time. Finally, the standard methods like particle collection on tape is still competitive with LIF. My question is: Did the authors try or are going to apply real-time aerosol data analysis? 2. L67 - principle or principal component analysis? 3. L116 – "The WIBS collects 3 channels of fluorescence intensity….." – collect channels or collects fluorescence intensity in 3 channels? 4. L170 – "…both saturating and non-fluorescent particles were retained…" – Did authors collect the particles? 5. L370 – "…gains…" or grains? 6. L494 - ..fluorescence and non-fluorescent particles.. - The phenomenon should not be compared with the property. 7. L 424 and further – I think that term "synthetic mixtures" for recorded numerical data is confusing and should be corrected. Firstly, it sounds like a chemical synthesis process. Secondly, the final result of clustering should be the same and independent whether the particle data are sorted or not. Otherwise, the order (sequence) of detected particles would change final result. I think that actual meaning of used data is well described in L298-300 ("...subset taken from the pool of particles.". 8. L 426 – "analytically synthesized" – analysis has opposite meaning to synthesis should be corrected 9. L 428, 431, 434, 436, – "…mixture synthesized…" – see point 7. 10. The authors compared clustering ability using selected small groups of substances. It would be interesting to see the clustering output for all 14 types together. Why it was not presented?

---

## Short Comment (SC1) · 30 Jun 2018

The study presented is an extremely well structured and written investigation into the use of Hierarchical Agglomerative Clustering for classification of biological aerosol using a UV-LIF sensor, and will make an excellent addition to the literature upon publication.

However, the authors may have made a small error [L161-L162] where they state that

the conclusions for Ruske et al. (2017) were for ambient data, whereas in the abstract they correctly state that the study was on standardised laboratory particles [L19-L20]. Please could you correct this prior to final publication.

In addition the authors may wish to consider the following comments prior to publication.

[L78-L79] Would it be possible to clarify the starting conditions for supervised learning you are referring to? Hyper-parameter selection is an extremely important consideration for neural networks, but other supervised techniques such as decision trees and ensemble methods do exist where low classification error can be attained without providing the algorithm with any initial conditions other than the training data.

[L84-L85] Is it necessary to apply unsupervised techniques to assess the advantages of supervised methods? Do you mean that supervised techniques require laboratory data of known types to assess their advantages? A very important disadvantage of supervised techniques is that they rely on adequate training data, and it is not clear at this point how much training data will be required to adequately represent an ambient environment, which is the point I think you are alluding to here.

[L186 - 187] Does the z-score rely on the assumption of normality? The z-scores of a normal random variable will be normally distributed whereas the z-scores of a non-normal random variable will be non-normally distributed. Applied to any data set, regardless of distribution, the resultant variables after z-scoring will have mean of 0 and standard deviation of 1. Is the purpose of standardising the data to prevent one of the variables from dominating in the analysis or to produce normally distributed data?

[L203] It would be worth noting that in Crawford et al., 2015, there are particles for which negative measurement of fluorescence was recorded. The option of log-transformations may have been overlooked, as the logarithm is undefined for negative values. This was not intended to imply an assumption of normality, although this assumption has been stated explicitly in Robinson et al., 2013. In these cases would you

recommend translating the fluorescence measurements to a range bounded below by 1, or alternatively would it be more appropriate to reject measurements for which the fluorescence produced was negative? It is also important to note that even if the data is log transformed, the data will still have a finite range due to the saturation point on the detector, and hence the data will have a truncated normal distribution rather than a normal distribution, and depending on how often saturation occurred there may still be a peak to the right hand side of the distribution. It is however, perfectly acceptable to apply HAC when the assumptions for best performance are not met as stated in Norusis, 2011.

[L222] How often did the CH index conclude that there were 2 clusters? When the CH index concluded a number of clusters other than 2, how much of an impact did this have on the quality of the results? Were the two cluster solutions always the best solution?

[L267-270 & Figure 3] The HAC algorithm may not necessarily output clusters in the same order that they were inputted as demonstrated in Figure 5. In Figure 3 for preparation strategy A for bacteria and diesel for the 80:20 ratio, is it possible to attain 80% misclassification for a two cluster solution? Perhaps I have misunderstood, but would this not mean that there were more diesel particles in the bacterial cluster and more bacterial particles in the diesel cluster, and hence a better classification error could be attained simply by swapping the labels on the clusters?

[Figure 3 & Table 2] Could you extend the results presented in Figure 3 to include at least one biological versus biological matchup? I notice when considering matching ups which contained only biological material the classification error is much higher. I believe that by not standardising the data this would cause the fluorescence to dominate more in the analysis. In the case of attempting to discriminate between fluorescent and non-fluorescent particles, this may be advantageous. However, in the case of attempting to discriminate between two different types of biological particle, it may be advantageous to give the size and shape measurements more weight, and

[Figure]

hence it would be better in these cases to standardise the data. In addition other instruments such as the WIBS-NEO will have fluorescence measurements over a much larger range and fluorescent measurements are recorded often above 10000. What would the implication then be when not standardising the data in this case?

———————————————

---

## Author Comment (AC1) · 13 Aug 2018

Public Comment- Simon Ruske (simon.ruske@student.manchester.ac.uk)

> Note regarding document formatting: black text shows original referee comment, blue text shows author response, and red text shows quoted manuscript text. Changes to manuscript text are shown as *italicized and underlined*. All line numbers refer to discussion/review manuscript.

[Public Comment] The study presented is an extremely well structured and written investigation into the use of Hierarchical Agglomerative Clustering for classification of biological aerosol using a UV-LIF sensor, and will make an excellent addition to the literature upon publication.

> [Author Response] Simon, thanks for taking the time to read and comment on the manuscript. We appreciate the useful comments, which will help improve the quality of the manuscript. We respond to each comment in detail below.

However, the authors may have made a small error [L161-L162] where they state that the conclusions for Ruske et al. (2017) were for ambient data, whereas in the abstract they correctly state that the study was on standardised laboratory particles [L19-L20]. Please could you correct this prior to final publication.

> I apologize for this mistake. I am not sure where this error came in our writing process, but I removed the incorrect statement, as requested: "Their conclusions, however, were based on ambient field data using unknown particle types and did not investigate laboratory generated particles of known origin."

In addition the authors may wish to consider the following comments prior to publication.
[L78-L79] Would it be possible to clarify the starting conditions for supervised learning you are referring to? Hyper-parameter selection is an extremely important consideration for neural networks, but other supervised techniques such as decision trees and ensemble methods do exist where low classification error can be attained without providing the algorithm with any initial conditions other than the training data.

> This may have been a bit of a miscommunication. We do not deal with any supervised learning methods in this manuscript. We trust your team as the experts in this area. Nicole simply wanted to provide a few sentences of general contrast between supervised and unsupervised methods. That is also why we pointed to your 2017 paper in this section. We have also included citation of your manuscript currently being reviewed in AMT.

[L84-L85] Is it necessary to apply unsupervised techniques to assess the advantages of supervised methods? Do you mean that supervised techniques require laboratory data of known types to assess their advantages? A very important disadvantage of supervised techniques is that they rely on adequate training data, and it is not clear at this point how much training data will be required to adequately represent an ambient environment, which is the point I think you are alluding to here.

> This is the way I understand some of the pros/cons of supervised and unsupervised. I agree that the community (probably you first) will continue to lean about how this all works together and how well lab-generated data can be useful to train supervised data algorithms. As you well know, the differences between nicely behaving lab particles and more complicated particles collected in the field confounds most areas of aerosol science to some degree. So these problems will not necessarily be trivial to solve, but I think collectively we are all learning little pieces that will help.

[L186 - 187] Does the z-score rely on the assumption of normality? The z-scores of a normal random
variable will be normally distributed whereas the z-scores of a non-normal random variable will be non-
normally distributed. **Applied to any data set, regardless of distribution, the resultant variables after**
**z-scoring will have mean of 0 and standard deviation of 1.** Is the purpose of standardising the data to
prevent one of the variables from dominating in the analysis or to produce normally distributed data?
Thanks to your prompting, we looked into these details and learned a bit more, which has been
helpful to us. You are right that the way we characterized the z-scoring process was not correct.
Talking back and forth with the university statistician, we now understand that values can indeed
be input scaled to a normal distribution or not. We chose to standardize our variables to a mean of
0 and a variance of 1 so that the output variables would be on comparable scales, but this is also
not the same as rigorously normalizing them in the rigorous sense. As a result, we have removed
the statement you correctly indicated was inaccurate and updated the sentence as follows:
Original text: "Standardization using the z-score method compares results to a normal (Gaussian)
population,
"
Updated text: "Standardization using the z-score method compares results to a normal (Gaussian)
population, and we have chosen to standardize our variables to a mean of 0 and a variance of 1 so
that the output variables would be on comparable scales."
[L203] It would be worth noting that in Crawford et al., 2015, there are particles for which negative
measurement of fluorescence was recorded. The option of logtransformations may have been overlooked,
as the logarithm is undefined for negative values. This was not intended to imply an assumption of
normality, although this assumption has been stated explicitly in Robinson et al., 2013. In these cases
would you recommend translating the fluorescence measurements to a range bounded below by 1, or
alternatively would it be more appropriate to reject measurements for which the fluorescence produced
was negative? It is also important to note that even if the data is log transformed, the data will still have a
finite range due to the saturation point on the detector, and hence the data will have a truncated normal
distribution rather than a normal distribution, and depending on how often saturation occurred there may
still be a peak to the right hand side of the distribution. It is however, perfectly acceptable to apply HAC
when the assumptions for best performance are not met as stated in Norusis, 2011.
My understanding is that negative fluorescence values can be observed after subtracting some
threshold value from the fluorescence intensity data. Instead of subtracting the data and looking
only at positive values, we did the same thing by filtering the data at several discreet thresholds.
This gets around the problem of negative values. In any case, we looked at three thresholding
scenarios (Table 3), i.e. no threshold, 3 sigma, and 9 sigma. The ultimate result is that we found
the most consistently positive results to be as a result of 3 sigma filtering, but this could be
different in other situations. You are correct about the fact that particles that exhibit saturation of
the detector in any channel will truncate a normal distribution.
[L222] How often did the CH index conclude that there were 2 clusters? When the CH index concluded a
number of clusters other than 2, how much of an impact did this have on the quality of the results? Were
the two cluster solutions always the best solution?
We did not explore solutions that had more than 2 solutions, simply as a matter of limited time.
There are certainly many scenarios in which individual bioparticle types (i.e. pollen, in many
instances) can split into two reasonable clusters by themselves, and so independently allowing 3

or more cluster solutions could significantly improve results in many cases. We just didn't have
the time to do this systematically, and so we chose to limit analysis to only 2 clusters in all cases.
To help clarify this point, we added text at:

L227: "In order to reduce the length and complexity of *discussion, analysis of results in Sections*
*4.1-4.3 was limited to using cluster products only from the 2-cluster solution. In some cases a 3-*
*cluster solution may have produced higher quality results, but these cases were not investigated.*"

[L267-270 & Figure 3] The HAC algorithm may not necessarily output clusters in the same order that
they were inputted as demonstrated in Figure 5. In Figure 3 for preparation strategy A for bacteria and
diesel for the 80:20 ratio, is it possible to attain 80% misclassification for a two cluster solution? Perhaps
I have misunderstood, but would this not mean that there were more diesel particles in the bacterial
cluster and more bacterial particles in the diesel cluster, and hence a better classification error could be
attained simply by swapping the labels on the clusters?

You are correct that the order of cluster numbering is unrelated to the order of particles input and
so the source of individual particles must be known already, but it is not possible to improve the
results by swapping labels in the way you suggest. We independently tracked the source of each
particle assigned to each cluster so we can rigorously calculate which particles were incorrectly
assigned. The numbering of the clusters is arbitrary and the naming was assigned simply as a
function of which particle was assigned in the largest concentration.

[Figure 3 & Table 2] Could you extend the results presented in Figure 3 to include at least one biological
versus biological matchup? I notice when considering matching ups which contained only biological
material the classification error is much higher. I believe that by not standardising the data this would
cause the fluorescence to dominate more in the analysis. In the case of attempting to discriminate between
fluorescent and non-fluorescent particles, this may be advantageous. However, in the case of attempting
to discriminate between two different types of biological particle, it may be advantageous to give the size
and shape measurements more weight, and hence it would be better in these cases to standardise the data.
In addition other instruments such as the WIBS-NEO will have fluorescence measurements over a much
larger range and fluorescent measurements are recorded often above 10000. What would the implication
then be when not standardising the data in this case?

This is another interesting idea, but it was beyond the scope of what we were able to accomplish
in the relatively short time we had available for this project. We chose to focus on the ability to
separate bio from non-bio particles. While we didn't explore all Scenarios (e.g. A-F) for
biological particles, we chose to look at bio-bio separations using Scenario B (i.e. Tables 2 and
3).

---

## Author Comment (AC2) · 13 Aug 2018

Note regarding document formatting: black text shows original referee comment, blue text shows author response, and red text shows quoted manuscript text. Changes to manuscript text are shown as *italicized and underlined*. Bracketed comment numbers (e.g. [R1.1]) were added for clarity. All line numbers refer to discussion/review manuscript.

[R1.0] This paper builds on existing literature examining unsupervised learning techniques to improve the interpretation and classification of data obtained with WIBS UV-LIF spectrometers. As shown in previous publications, Hierarchical Agglomerative Clustering (HAC) can serve as a robust data analysis method for classification/interpretation of bioaerosol data but the accuracy of technique is highly sensitive to the choice of clustering linkage and data pre-treatment (e.g., Crawford et al., 2015); this is further explored in this paper which elucidates how data pre-treatment choices such as choice of fluorescent threshold and log normalising data may influence clustering accuracy using laboratory samples of known particle types (Savage et al., 2017) in various synthetic mixtures, and thus the authors present tentative recommendations of data pretreatment regimes depending on the analysis goals. Overall the paper is well written and the computational experiments well thought out. The findings here are useful and further validate the usefulness of Hierarchical Agglomerative Clustering for interpretation of WIBS data. The results also provide a useful framework for testing Hierarchical Agglomerative Clustering data pre-treatment regimes for other atmospheric science data problems and neatly demonstrate the potential pitfalls of not rigorously performing such tests. I recommend publication after the following comments have been addressed.

[A1.0] Author response: We thank the referee for her/his positive summary of the manuscript and recommendation to publish after comments are addressed.

Specific comments
[R1.1] L73-77: The authors have conflated some of the terminology relating to unsupervised and supervised leaning methods. I'm uncomfortable with the use of the term clustering when discussing supervised methods as clustering specifically relates to cluster analysis. I suggest replacing "clustering techniques" with "classification algorithms" and "(trains) the clustering algorithm" with "(trains) the classification algorithm".

[A1.1] The referee raises a good point. We changed terminology on page 2 according the referee suggestions, as listed below:
- L68: "*Classification algorithms, including several* clustering techniques in particular, have shown successful results …"
- L73: " *Classification algorithms* can be divided …"
- L76: "This type of method enhances (trains) the  *classification* algorithm in that the output  *groups* are predetermined …"

[R1.2] L120: Please state the bands and what they relate to.

[A1.2] Additional text was added, as shown below:
"The WIBS collects 3 channels of fluorescence intensity information (FL1, FL2, and FL3), particle size, and particle asymmetry for each interrogated particle. *The bands of excitation and fluorescence emission are: FL1 ($\lambda_{ex} = 280$ nm, $\lambda_{em} = 310 – 400$ nm), FL2 ($\lambda_{ex} = 280$ nm, $\lambda_{em} = 420 – 650$ nm), and FL3 ($\lambda_{ex} = 370$ nm, $\lambda_{em} = 420 – 650$ nm).* The excitation and emission wavelengths chosen for each of the 3 fluorescence channels were designed to maximize the information gained about key biological fluorophores present in a broad range of bioparticles (Kaye et al., 2005; Pöhlker et al., 2012). *Early generations of UV-LIF bioaerosol spectrometers were often interpreted to be able to detect proteins via channels similar to FL1 and products of active cellular metabolism (i.e. riboflavin and NAD(P)H) via channels similar to FL3, but these approximations are gross simplifications that confound more detailed investigation of particle types.*"

[R1.3] L198: Can the authors please clarify why they have used log spaced bins. Do you mean that you have taken a log of the data and it is binned naturally by the discrete nature of the detector resolution (i.e., fine bins) or have you binned the data into specific (coarse) log bins? If it is the latter can you please state what the bins are and can you comment on how forcing the data to in bins may influence the clustering? My concern here is that too coarsely binning the data may create artificial hotspots due to reduced resolution and bias the clustering, reducing the capacity to differentiate between particles with similar properties. Can the authors comment on this and demonstrate the effect this may have by providing an example for comparison where the data is converted to log space and not binned. I also wonder if the bins should be normalised by the bin width to further complicate matters.

[A1.3] Aspects of this discussion are presented in L209-212. To summarize in different words, the data values from a given channel were either used as recorded (i.e. "value") or as logarithmically transformed (i.e. "log(value)"), depending on the Scenario. The values were not forced into specific bins, but rather input into the cluster algorithm using the exact value in either of these forms. The reason that logged values can provide different results by HAC is that the distance between points is different in linear space or log space, because the cluster process does not independently take into account whether a value is as recorded or as log(value). Because many real-world particle variables can present normal distributions only in log space (i.e. lognormal size distributions), we explored inputting values in both raw and log forms.

The following sentence was added to the manuscript at L211 for clarity:
"*By this process, data values were input into the algorithm as log(value), but without additional binning*."

[R1.4] L254: Can the authors comment on the environmental applicability of the chosen ratios. I would suspect that in an urban environment you may expect something closer to a ratio of 1:99 fungal to diesel particles with the converse being true in a forest environment. How does the clustering perform under such extreme mismatches?

[A1.4] We originally explored three different ratios of particle concentrations (80:20, 50:50, and 20:80) for each of the three match-ups discussed in Figure 3 in order to show that input ratio can be important to how the algorithm responds. This was certainly not intended to be exhaustive, and one could additionally explore more extreme ratios. So to limit the scope of the analysis here, we chose to present evidence only that the ratio matters, without trying in all cases to predict ratios that could be relevant to a wider range of ambient environments.

The question the referee brings up is interesting, however, and so we explored 1:99 ratios of each of the three particle type combinations presented in Figure 3, where the bioparticle is the minority concentration in each experiment. The results are shown below in a plot/table form identical to how they are presented in Figure 5. The Bacteria:Diesel and Fungi:Dust separations still performed quite well (6.6% and 13.5% misclassification, respectively), even with the extreme mismatch of input concentrations. The Fungi:Diesel separation was poor, however, in a 2-factor solution, because the Diesel particles split into both clusters, and the Fungi particles were likely too low in concentration to influence the cluster properties. We added text including a summary
of these new experiments to the manuscript at L304:
*"To extend the investigation of particle input ratio, the three match-ups presented in Figure 3*
*were investigated using Scenario B with 1% bioparticles and 99% non-bioparticles in each*
*respective case. In these experiments the Bacteria:Diesel and Fungi:Dust particles separated*
*relatively well (6.6% and 13.5% misclassification, respectively). The Fungi:Diesel separation*
*was poor, however, because the Diesel particles were nearly evenly split into both clusters, and*
*the Fungi particles were too low in concentration to influence the cluster properties. More*
*investigation is needed to explore how extreme disparities in particle ratio could negatively*
*influence cluster quality in real-world settings."*

**Part A: Individual Clusters** (Particle Number)

**Fungi : Diesel**

| Cluster | B3 | F2 | S4 | D12 |
|---|---|---|---|---|
| 1 | - | 37 | 2588 | - |
| 2 | - | 0 | 1111 | - |

**Part B: Grouped Clusters** (Particle Number)

| Cluster | Bio | Non-bio |
|---|---|---|
| 1 | 37 | 2588 |
| 2 | 0 | 1111 |

**Part C: Summary** (Cluster Quality)

| Total P. | Miscl. | Cat. |
|---|---|---|
| 2625 | 98.6% | Fungi |
| 1111 | 0.0% | Diesel |

**Bacteria : Diesel**

| Cluster | B3 | F2 | S4 | D12 |
|---|---|---|---|---|
| 1 | 57 | - | 4 | - |
| 2 | 0 | - | 5653 | - |

| Cluster | Bio | Non-bio |
|---|---|---|
| 1 | 57 | 4 |
| 2 | 0 | 5653 |

| Total P. | Miscl. | Cat. |
|---|---|---|
| 61 | 6.6% | Bacteria |
| 5653 | 0.0% | Diesel |

**Fungi : Dust**

| Cluster | B3 | F2 | S4 | D12 |
|---|---|---|---|---|
| 1 | - | 45 | - | 7 |
| 2 | - | 12 | - | 5650 |

| Cluster | Bio | Non-bio |
|---|---|---|
| 1 | 45 | 7 |
| 2 | 12 | 5650 |

| Total P. | Miscl. | Cat. |
|---|---|---|
| 52 | 13.5% | Fungi |
| 5662 | 0.2% | Dust |

[R1.5] L238: Would it be possible to show examples of the cluster centroids for a case where there is
significant misclassification? This may illuminate why the algorithm is failing to correctly attribute
particles. It may also be useful to examine the fluorescence/AF characteristics of each cluster as a
function of size here. A 2D histogram or color density plot could show distinct hot spots that haven't been
separated correctly and could provide a basis for manual separation based on sensible thresholds.
[A1.5] To address the referee's suggestion, we included an additional set of plots here as
suggested. The results below correspond to the match-up between Bacteria 1 and Bacteria 3 using
Scenario B and the 3-sigma threshold, which corresponds to Experiment 22 from Table 2 (65%
misclassification). The two colors of dots in the plots represent clusters 1 and 2. In this case it is
still unclear how to utilize a single threshold to separate between the two particle types here.
In the process of analyzing results of this study we produced countless plots and tables, each of
which showed slightly different angles of the same story. We chose to simplify the results in
many cases to make the manuscript shorter and more manageably readable. We find that the table
of fluorescence intensity and AF median values (Table 2 from original data published in Savage
et al., 2017) often summarizes the differences in the particle types rather well and so were rarely
able to separate using 2D histograms as the referee suggests. One example of these two additional
plots is included here for reference, however.

[Figure]

[R1.6] L312-315: Can you describe the method for producing the soot as they seem rather large as
compared to that in the study of Toprak and Schnaiter (2013) which were also coincidently found to be
weakly fluorescent in FL1. Perhaps the soot used in this study is larger and more fluorescent than we may expect of ambient/urban soot which may cause some of the difficulty in correctly attributing in in some
cases?
[A1.6] The method for aerosolization of particle types discussed was presented in Section 3.2 of
the associated Savage et al., 2017. Specifically, the aerosolization details related to soot are
copied here:
From Page 4284, Section 3.2.3 of Savage et al., 2017: "Dry powders were aerosolized by
mechanically agitating material by one of several methods mentioned below and passing
filtered air across a vial containing the powder. For each method, approximately 2.5–5.0
g of sample was placed in a 10 mL glass vial. For most samples (method P1), a stir bar
was added, and the vial was placed on a magnetic stir plate. Two tubes were connected
through the lid of the vial. The first tube connected a filter, allowing particle-free air to
enter the vessel. The second tube connected the vial through approximately 33 cm of
conductive tubing (0.25 in. inner diam.) to the WIBS for sample collection."
The referee is correct that the method of producing/aerosolizing particles, including soot, will
bear heavily on the fluorescent properties observed. In particular, different aerosolization
methods are likely to produce very different size distributions, which then will dictate the overall
fluorescence properties. For this reason, we included the following statements in the Savage et al.,
2017 paper:
From Page 4292, Section 4.3: "It is important to note, however, that the method chosen
for particle generation in the laboratory strongly impacts the size distribution of
aerosolized particles. For example, higher concentrations of an aqueous suspension of
particle material generally produce larger particles, and the mechanical force used to
agitate powders or aerosolize bacteria can have strong influences on particle viability and
physical agglomeration or fragmentation of the aerosol (Mainelis et al., 2005). So, while
the absolute size of particles shown here is not a key message, the relative fluorescence at
a given size can be informative."
The referee points out that the work by Toprak and Schnaiter (2013) presented small soot
particles that also exhibited relatively weaker fluorescence in FL1. This is consistent with the
expectation that fluorescence intensity will scale strongly with particle size. Differences in
particle size could also impact clustering separation properties somewhat, and so further
investigation of clustering using multiple narrow size ranges of different types of particles could
further explore this process. This exhaustive process was beyond the scope of this work, however.
To make sure these points are clear in the revised manuscript we have added the following text at
L327:
*"It is also important to note here that the method of aerosolization for each particle type plays an*
*important role in the observed size distribution and so results involving laboratory particles*
*should be interpreted with this in mind. Observed fluorescence properties, in contrast, are*
*expected to be conserved at a given particle size and intrinsically related to particle*
*composition."*
[R1.7] L384: Would we expect to be able to differentiate between 2 different particles of the same type
with such coarse spectral resolution?
[A1.7] The referee's implied point is correct. No, we would not expect to be able to separate
between very similar types of particles using such coarse resolution as is available in the WIBS.

Frankly, the fact that HAC paired with WIBS data was able to separate as well as it did was somewhat remarkable and surprising. To make the point clearer, we added text at the end of that paragraph as follows at L390:

"…separating more finely to quantify differences between types of individual biological particles is  significantly more challenging *and not likely to be possible in most situations.*"

[R1.8] L415: Again I wonder if the use of too coarsely separated bins may compromise the 9-sigma thresholding and cause misclassification?

[A1.8] This question also loops back to [R1.3] and stems from a miscommunication. Values of the five WIBS data parameters were not separately binned (either during the logging process or when used as recorded), but are input into the cluster algorithm in the same spacing provided in the raw output of the instrument. The bin resolution is therefore limited by the WIBS optics and PMT settings.

Further, fluorescence intensity is relayed by a integer units between 0 and 2047, and resolution is not a limiting factor. For example, see Figure 5 of the Savage et al. 2017 paper. Biological particles typically exhibit median fluorescence intensity much higher than non-biological particles, thus using different threshold strategies can help separate particle classes from one another by this strategy.

[R1.9] L514: Can the authors comment on the applicability of their findings to new high resolution UV-LIF instruments that are beginning to become commercially available. Some of these new instruments have significantly more channels/greater fluorescent resolution than the WIBS.

[A1.9] This is a helpful suggestion. To extend the applicability of results, the text was amended as follows:
"Results here are  generally extendable to other UV-LIF instruments, *however, whether they offer single or many channels of emission spectral resolution, in that the methods of particle pre-preparation and the impact of particle number ratio are likely to relay similar effects on clustering strategy."*

[R1.10] Technical corrections
L63: instruments, not instrument.
L370: grains, not gains.
L112: Suggest "Experimental and Computational Methods"
L131: "each of the three"
L181: "was the best"

[A1.10] All typos corrected.

---

## Author Comment (AC3) · 13 Aug 2018

> Note regarding document formatting: black text shows original referee comment, blue text shows author response, and red text shows quoted manuscript text. Changes to manuscript text are shown as *italicized and underlined*. Bracketed comment numbers (e.g. [R1.1]) were added for clarity. All line numbers refer to discussion/review manuscript.

[R2.0] This manuscript discusses application of Hierarchical Agglomerative Clustering (HAC) to analysis of data collected using the Wideband Integrated Bioaerosol Sensor (WIBS4A). While real-time detection of bioaerosols has been quite well controlled, the analysis and classification is still challenging and vital problem. Therefore, investigation and improvements in this area are very important and crucial for understanding the abilities and limitations of LIF aerosol detectors. The manuscript is well written and in detail reveals important problems of fluorescence data analysis of bioaerosols. I recommend presented manuscript to publication, however some corrections and further explanations to the following remarks will be appreciated:

> [A2.0] Author response: We thank the referee for her/his positive summary of the manuscript and recommendation to publish after comments are addressed.

[R2.1] 1. The techniques of single particle detection using LIF devices, like WIBS, reached relatively high reliability and perfection. The device collects data in real time, on the other hand the presented results are offline. The data analysis takes a long time. Finally, the standard methods like particle collection on tape is still competitive with LIF. My question is: Did the authors try or are going to apply real-time aerosol data analysis?

> [A2.1] I think the statement that "LIF devices … reached relatively high reliability and perfection" is already an very optimistic statement, but I agree that when operated and analyzed properly the data can often be useful. The referee's suggestion about real-time data analysis is an interesting idea that has been discussed. We are working on this type of analysis from a different angle and with respect to a different class of instruments, but we have not had the ability to investigate real-time analysis strategies with respect to WIBS data. This would be a worthwhile project, but is outside the scope of what we were aiming to accomplish in this study and would likely require dedicated project funding.

[R2.2] 2. L67 - principle or principal component analysis?

> [A2.2] In this case the word "principal" is the correct one. I often get this word confused with "principle" and have to look up the definitions to make sure I'm correct.

[R2.3] 3. L116 – "The WIBS collects 3 channels of fluorescence intensity. . ..." – collect channels or collects fluorescence intensity in 3 channels?

> [A2.3] This was indeed poor grammatical construction. The sentence has been changed to: "The WIBS collects *information about*  fluorescence intensity  *in three channels* …"

[R2.4] 4. L170 – ". . .both saturating and non-fluorescent particles were retained. . ." – Did authors collect the particles?

[A2.4] We did not physically collect the particles. The wording here was unfortunately confusing. In this case we have "retained" the data in the analysis process by not removing particles based on certain attributes. To clarify, the word "retained" was changed to "analyzed" as shown here: "*... both saturating and non-fluorescent particles were analyzed* retained …"

[R2.5] 5. L370 – ". . .gains. . ." or grains?

[A2.5] This is a typo; "gains" was corrected to "*grains*".

[R2.6] 6. L494 - ..fluorescence and non-fluorescent particles.. - The phenomenon should not be compared with the property.

[A2.6] This typo was changed for the discussion version of the manuscript to be "fluorescent and non-fluorescent particles."

[R2.7] 7. L 424 and further – I think that term "synthetic mixtures" for recorded numerical data is confusing and should be corrected. Firstly, it sounds like a chemical synthesis process. Secondly, the final result of clustering should be the same and independent whether the particle data are sorted or not. Otherwise, the order (sequence) of detected particles would change final result. I think that actual meaning of used data is well described in L298-300 ("...subset taken from the pool of particles..".

[A2.7] The term "synthetic mixtures" is indeed confusing terminology, and this is a point raised also by Referee #3 (i.e. [R3.1], [R3.3], and [R3.6]). Referee #3 suggested the term "computational simulations" or "simulated mixtures" among several possibilities, and we have changed the text in a variety of places through-out the manuscript to reflect this new terminology.

[R2.8] 8. L 426 – "analytically synthesized" – analysis has opposite meaning to synthesis should be corrected

[A2.8] Here the term was changed to "computationally simulated."

[R2.9] 9. L 428, 431, 434, 436, – ". . .mixture synthesized. . ." – see point 7.

[A2.9] The word "synthesized" was changed to "simulated" in each of these cases and all others within the manuscript.

[R2.10] 10. The authors compared clustering ability using selected small groups of substances. It would be interesting to see the clustering output for all 14 types together. Why it was not presented?

[A2.10] This additional experiment might be interesting, but it is unlikely to add anything to the general nature of the conclusions. The 14 types of particles assembled for these match-up experiments (i.e. Sections 4.1 – 4.3) were meant to be individually instructive, but not to represent the entirety of the types of particles one might see in a more complex, ambient environment. So collecting all 14 into one experiment would represent another experimental combination, but would in itself not be any more relevant than the individual simulations already discussed.

---

## Author Comment (AC4) · 13 Aug 2018

> Note regarding document formatting: black text shows original referee comment, blue text shows author response, and red text shows quoted manuscript text. Changes to manuscript text are shown as *italicized and underlined*. Bracketed comment numbers (e.g. [R1.1]) were added for clarity. All line numbers refer to discussion/review manuscript.

[R3.0] This paper describes methods and results which should help improve the interpretation and use of data obtained with UV-LIF instruments such as the WIBS. The WIBS measures light scattering, a light scattering asymmetry factor, and fluorescence in three channels. Fielded instruments with data rates that can exceed hundreds of particles per minute are available. This paper uses a large set of WIBS data measured for individual materials (Savage et al. 2017) to evaluate different preprocessing procedures for analysis of such data. Mathematical simulations of externally mixed particles of known composition are studied. The findings should be useful not only for understanding WIBS data, but more broadly in applying Hierarchical Agglomerative Clustering to some other problems in aerosol analytical chemistry. I recommend publication. However, I request that several confusing items be made less confusing.

> [A3.0] Author response: We thank the referee for her/his positive summary of the manuscript and recommendation to publish after comments are addressed.

[R3.1] The use of the term "synthetic mixtures" (L31-32, L424, 707, L734) is confusing. Chamber studies with synthetic mixtures of real aerosols and real gases are not uncommon in aerosol science. A google search of "synthetic mixture" provides discussions of various real "synthetic mixtures." I only looked at the first 8 or so items in that search, but I saw none with the meaning used in this paper. The online dictionaries I saw do not indicate this use of "synthetic" (which as far as I can tell indicates something about numerical or computational). Synthetic organic chemists make real chemicals. If "synthetic mixtures" is used for the simulated data investigated here, what terminology is left for researchers to use when they make real synthetic mixtures of aerosols in a chamber and investigate changes in clusters as time passes and as particles agglomerate? I do not see how a reader can see from the abstract or even well into this paper that "synthetic" is being used in this highly non-standard way, and that Savage et al., 2017 did not measure mixtures of particles. The "synthetic mixtures" are actually numerical (or mathematical) simulations of the WIBS the data that should be obtained for dilute mixtures of particles. Real mixtures of particles can form agglomerates, and some may agglomerate quickly unless they are sufficiently dilute.

> [A3.1] This is a good point that we had not previously considered. The same point was raised by Referee #2 [R2.7, R.2.8, and R2.9]. We removed all use of the term "synthetic mixtures" and changed most instances of the term to "simulated mixtures." Note that this comment also impacts comments [R3.3] and [R3.6].

[R3.2] L 20-22 (Abstract). "Here we show for the first time a systematic application of HAC to a comprehensive set of laboratory data collected using the wideband integrated bioaerosol sensor (WIBS-4A) (Savage et al., 2017)." Suggest change to: "Here we show for the first time a systematic application of HAC to a comprehensive set of laboratory data collected for individual particle types using the wideband integrated bioaerosol sensor (WIBS-4A) (Savage et al., 2017). Here the WIBS data for single-composition aerosols is combined numerically to generate data to simulate WIBS values for mixtures of aerosol."

> [A3.2] The text of the abstract was modified as suggested.

[R3.3] L31-32 (Abstract): "Lastly, six synthetic mixtures of four to seven components were analyzed."
Might be changed to: "Numerical simulations of mixtures of four to seven components were HAC
analyzed."

[A3.3] The text of the abstract was changed as requested to:
"Lastly, six *numerical simulations of*  mixtures of four to seven components were
analyzed *using HAC*."

[R3.4] L424: "Investigating cluster ability to separate complex synthetic mixtures" Might be changed to:
Investigating the capability to separate particles in simulations of complex synthetic mixtures

[A3.4] The sub-title was changed along the suggested lines to:
"Investigating *the capability*  to separate *particles in simulations of* complex
 mixtures"

[R3.5] L426-429: "To better simulate real-world scenarios, we analytically synthesized six mixtures of
particles by pooling existing data from selected particle types in prescribed ratios. Each mixture was
synthesized to roughly represent a different hypothetical mixture of particles that might be expected."
"Analytically" suggests equations or functions were used in obtaining the data for the mixtures. Isn't
"numerically" or "computationally" what is meant?

[A3.5] The word "analytically" was changed to "computationally."

[R3.6] L426-429 might be changed to: "To better simulate real-world scenarios, we numerically
simulated six mixtures of particles by pooling existing WIBS data from selected particle types in
prescribed ratios. Each simulated mixture was assembled to roughly represent a different hypothetical
mixture of particles that might be expected. Also, the particles in each simulated mixture are assumed to
be so dilute that any agglomeration is negligible. " Also, a significant fraction of readers read the abstract
and then look at the figures to see what the results will be. Adding clarifying words to the figure captions
and tables would be useful.

[A3.6] These are good suggestions that add clarity to the text. The section was re-written with the
suggested text. Words "computational" or "numerical" added to captions of several figures and
tables to increase clarity, as suggested.

[R3.7] [a] I don't know what "normalized to particle size" means here. Please clarify, possibly with an
equation. Please also give the ranges of error in particle sizes expected. [b] Why is scenario D worse than
B? I think it is because D adds noise to the FL signals, making them less informative by decreasing the
S/N. This added noise occurs in the elastic scattering measurements, and also results from the
approximations used in estimating solutions to the inverse problem for size (with unknown shape,
orientation and refractive index). If the scattering measurement and the solution to the inverse problem
were perfect, then D and B should give very similar results, at least for spherical particles and some
methods of normalizing to particle size and shape. It may be useful to cite a paper or data with WIBS
measurements of size and fluorescence for uniformly-sized fluorescent PSL. For a single size of PSL, do
plots of the WIBS-measured scattering and fluorescence fall on a line or are they spread more randomly?
Even for a spherical PSL particle, with known refractive index, would you suspect that the noise is large
enough to make D less useful than B?

[A3.7] To clarify the first question [a], additional text was added to L207:
"…fluorescence intensity was normalized to particle size *(by dividing fluorescence intensity value
by light scattering signal when a particle interacts with the diode laser beam)* in order to …"

102

With respect to the second question [b], the referee is likely correct that results for Scenario D (fluorescence normalized) are worse than for Scenario B (fluorescence not normalized), because for Scenario D additional uncertainty with respect to size is propagated into the intensity value. Normalizing in this way would also propagate uncertainty for field measurements, and so given the poorer results of the tests analyses represented here we chose not to further explore parameters represented by Scenario D.

[R3.8] Can the authors say anything about the length of times bacteria or fungal spores might last in an urban environment before a significant fraction of the bioparticles combine with soot, and how that might affect the usefulness of the WIBS? I'll be very interested to see the results when (sometime in the future) the authors inject bacteria or fungal spores into a chamber, add soot particles, use the WIBS to sample with time, and then repeat the some of the analyses in this paper with the results given as a function of time.

[A3.8] This an interesting question, but we do not have a good answer to the hypothetical thought about atmospheric lifetimes of these particles at this point. It would be great to explore external mixing of different particles types in the future in order to see how these mixtures could further influence fluorescence and particle size properties observed by instruments like the WIBS. This is beyond the scope of the experimental process for now.

[R3.9] L23: In abstract: "ratio" of what? In the text, "ratio" first appears in "distance ratio." Suggest change first use of "ratio" in abstract to "ratio of particle concentrations."

[A3.9] Text edited as requested.

[R3.10] L117: please add wavelength ranges of FL1 to FL3. Aim for a little broader set of readers.

[A3.10] This was also requested by Referee #1. Additional text was added, as shown here: "The WIBS collects 3 channels of fluorescence intensity information (FL1, FL2, and FL3), particle size, and particle asymmetry for each interrogated particle. *The bands of excitation and fluorescence emission are: FL1 ($\lambda_{ex}$ = 280 nm, $\lambda_{em}$ = 310 – 400 nm), FL2 ($\lambda_{ex}$ = 280 nm, $\lambda_{em}$ = 420 – 650 nm), and FL3 ($\lambda_{ex}$ = 370 nm, $\lambda_{em}$ = 420 – 650 nm)."*

[R3.11] L171: replace "will be" with "were".

[A3.11] The phrase "will be" changed to "*is*" to match correct tense.

[R3.12] L199: Suggest change to: Ambient particle number vs size distributions can often be well approximated by lognormal distributions (citation), although specific subsets of particles, such as bacteria, pollens or fungal spores, may not exhibit lognormal distributions.

[A3.12] Text revised as suggested.

[R3.13] L245: "placed into a conceptual pool"? How about, "A subset of the particles were selected randomly for analysis"?

[A3.13] Text was changed, as suggested, to:
"For each trial, a *subset*  of particles from each material type was *selected randomly for HAC analysis* ."

153
154 [R3.14] L258-259: "diesel soot particles . . . commonly observed . . ." Is this referring to WIBS
155 measurements? Please provide a citation(s).
156
157 [A3.14] The text as originally written was indeed over-stated and confusing. The text has been
158 revised to the following:
159 "The first two trials include diesel soot particles, because *light-absorbing carbon aerosol*  are
160 commonly observed in  *aerosol*  samples with  anthropogenic
161 influence *(Bond et al., 2013)* …"
162
163 [R3.15] L299-300: Do you mean: "In each case the input particles are a random subset . . ."
164
165 [A3.15] Yes, the words "number of" was inserted incorrectly here and the typo was corrected as
166 suggested by the referee.